

# Regional and seasonal truncation errors of trajectory calculations using ECMWF high-resolution operational analyses and forecasts

Thomas Rößler[1,2], Olaf Stein[1], Yi Heng[3], and Lars Hoffmann[1]

[1]Forschungszentrum Jülich, Jülich Supercomputing Centre, Jülich, Germany
[2]now at: Department of Mathematics, University of Wisconsin–Milwaukee, Milwaukee, Wisconsin, United States
[3]School of Chemical Engineering and Technology, Sun Yat-sen University, Guangzhou, China

*Correspondence to:* Thomas Rößler (t.roessler@fz-juelich.de)

**Abstract.** Lagrangian particle dispersion models (LPDMs) are indispensable tools to study atmospheric transport processes. The accuracy of trajectory calculations, which form an essential part of LPDM simulations, depends on various factors. Here we focus on truncation errors that originate from the use of numerical integration schemes to solve the kinematic equation of motion. The optimization of numerical integration schemes to minimize truncation errors and to maximize computational speed is of great interest regarding the computational efficiency of large-scale LPDM simulations. In this study we analyzed truncation errors of six explicit integration schemes of the Runge Kutta family, which we implemented in the Massive-Parallel Trajectory Calculations (MPTRAC) model. The simulations were driven by wind fields of the latest operational analysis and forecasts of the European Centre for Medium-range Weather Forecasts (ECMWF) at T1279L137 spatial resolution and 3 h temporal sampling. We defined separate test cases for 15 distinct domains of the atmosphere, covering the polar regions, the mid-latitudes, and the tropics in the free troposphere, in the upper troposphere and lower stratosphere (UT/LS) region, and in the lower and mid stratosphere. For each domain we performed simulations for the months of January, April, July, and October for the years of 2014 and 2015. In total more than 5000 different transport simulations were performed. We quantified the accuracy of the trajectories by calculating transport deviations with respect to reference simulations using a 4th-order Runge-Kutta integration scheme with a sufficiently fine time step. We assessed the transport deviations with respect to error limits based on turbulent diffusion. Independent of the numerical scheme, the truncation errors vary significantly between the different domains and seasons. Especially the differences in altitude stand out. Horizontal transport deviations in the stratosphere are typically an order of magnitude smaller compared with the free troposphere. We found that the truncation errors of the six numerical schemes fall into three distinct groups, which mostly depend on the numerical order of the scheme. Schemes of the same order differ little in accuracy, but some methods need less computational time, which gives them an advantage in efficiency. The selection of the integration scheme and the appropriate time step should possibly take into account the typical altitude ranges as well as the total length of the simulations to achieve the most efficient simulations. However, trying to generalize, we recommend the 3rd-order Runge Kutta method with a time step of 170 s or the midpoint scheme with a time step of 100 s for efficient simulations of up to 10 days time based on ECMWF's high-resolution meteorological data.



# 1 Introduction

Lagrangian particle dispersion models (LPDMs) have proven to be useful for understanding the properties of atmospheric flows, particularly for problems related to transport, dispersion, and mixing of tracers and other atmospheric properties (e. g. Lin et al., 2012; Bowman et al., 2013). Commonly used LPDMs include the Flexible Particle (FLEXPART) model (Stohl et al., 2005), the Hybrid Single-Particle Lagrangian Integrated Trajectory (HYSPLIT) model (Draxler and Hess, 1998), the Lagrangian Analysis Tool (LAGRANTO) (Wernli and Davies, 1997; Sprenger and Wernli, 2015), the Numerical Atmospheric-dispersion Modelling Environment (NAME) (Jones et al., 2007), and the Stochastic Time-Inverted Lagrangian Transport (STILT) model (Lin et al., 2003). While all these models are applied to solve similar tasks, they differ in specific choices such as the numerical methods or vertical coordinates that are used. In this study we apply the rather new model Massive-Parallel Trajectory Calculations (MPTRAC) (Hoffmann et al., 2016), which was recently developed at the Jülich Supercomputing Centre, Germany. MPTRAC was primarily designed to conduct trajectory calculations for large-scale simulations on massive-parallel computing architectures. Computational efficiency is an important aspect during the development of such a model.

LPDMs simulate transport and diffusion of atmospheric tracers based on trajectory calculations for many air parcels that move with the fluid flow in the atmosphere. The accuracy of these calculations has been the subject of numerous studies (e. g., Kuo et al., 1985; Rolph and Draxler, 1990; Seibert, 1993; Stohl et al., 1995; Stohl and Seibert, 1998; Stohl et al., 2001; Davis and Dacre, 2009). According to reviews of Stohl (1998) and Bowman et al. (2013), trajectory calculations have errors that arise from three sources: (i) errors in the gridded winds themselves, which could result from measurement error that enter the analyzed fields through the data assimilation process or from Eulerian model approximations, such as subgrid-scale parameterizations; (ii) sampling errors that follow from the fact that velocity fields are available only at finite spatial and temporal resolution and must be interpolated to particle locations; and (iii) truncation errors that originate from the use of an approximate numerical scheme to integrate the kinematic equation of motion in time. Bowman et al. (2013) point out that (i) and (ii) are usually the limiting factors for the accuracy of trajectory calculations, whereas high numerical accuracy and significant reduction of truncation errors can be achieved by reducing the size of the time step of the numerical integration scheme. The size of the time step is usually the most important factor that controls the trade-off between numerical accuracy and computation time. However, it needs to be stressed that appropriate selection of the numerical scheme and optimization of the size of the time step is still mandatory to maximize computational efficiency. This is particular important for large-scale simulations, like Lagrangian transport simulations aiming at emission estimation by means of inverse modeling (e. g. Stohl et al., 2011; Heng et al., 2016) or long-term simulations coupled to chemistry climate models (e. g. Hoppe et al., 2014).

In the following we present an assessment of six numerical integration schemes, all belonging to the class of explicit Runge-Kutta methods (e. g., Press et al., 2002; Butcher, 2008), for atmospheric trajectory calculations. Seibert (1993) studied the truncation errors of some of these schemes based on analytic flow types such as purely rotational flow, purely deformational flow, wave flow, and accelerated deformational flow. Here we decided to focus on tests with realistic wind fields obtained from high-resolution operational analyses and forecasts provided by the European Centre for Medium-Range Weather Forecasts (ECMWF). The T1279L137 ECMWF operational analysis data used here have 16 km effective horizontal resolution, about



$180 - 750\,\mathrm{m}$ vertical resolution at $2 - 32\,\mathrm{km}$ altitude, and are provided at $3\,\mathrm{h}$ synoptic time intervals. We estimated the truncation errors of the numerical methods for 5 latitude bands and 3 altitude ranges of the atmosphere, covering the free troposphere, the upper troposphere and lower stratosphere (UT/LS) region, and the mid stratosphere. We studied the seasonal and inter-annual variability of the truncation errors for the years 2014 and 2015. We systematically assessed trade-offs between accuracy and

computation time to infer the computational efficiency of the integration methods. The results of our study will be transferable to most other currently used LPDMs. Using most recent meteorological data, the results will be of interest for many current and future LPDM studies using this new data set.

In Sect. 2 we present the Lagrangian particle dispersion model MPTRAC together with an overview on the meteorological data. The selected numerical integration schemes and the diagnostic variables are introduced and the experimental set-up is

described. Section 3 shows transport deviations from case studies followed by a general analysis of the error behavior in terms of seasonal and regional characteristics. Scalability and performance on a high-performance computing system are discussed. In Sect. 4 we conclude with suggestions for best-suited integration schemes and optimal time step choice in order to achieve most effective simulations of large-scale problems on current high-performance computing systems.

## 2 Methods and Data

### 2.1 Lagrangian particle dispersion model

In this study we apply the Lagrangian particle dispersion model MPTRAC (Hoffmann et al., 2016) to conduct trajectory calculations. MPTRAC has been developed to support the analysis of atmospheric transport processes in the free troposphere and stratosphere. In recent studies it has been used to perform transport simulations for volcanic eruptions and to reconstruct time- and height-resolved emission rates for these events (Heng et al., 2016; Hoffmann et al., 2016). The primary task of

MPTRAC is to solve the kinematic equation of motion for atmospheric air parcels. It calculates air parcel trajectories based on given meteorological wind fields. Turbulent diffusion and subgrid-scale wind fluctuations are simulated following the approach of the FLEXPART model (Stohl et al., 2005). Additional but unused modules can simulate the sedimentation of air parcels and the decay of particle mass. The model is particularly suited for ensemble simulations on supercomputers due to its efficient Message Passing Interface (MPI) / Open Multi-Processing (OpenMP) hybrid parallelization.

### 2.2 ECMWF operational analysis

Air parcel transport in MPTRAC is driven by meteorological wind fields. In principle any gridded data produced by general circulation models, atmospheric reanalyses, or operational analyses and forecasts can be used for this purpose. Reanalyses and forecasts benefit from well-established meteorological data assimilation methods (Rabier et al., 2000; Buizza et al., 2005) which help to better constrain the modelled circulation fields to reality. While atmospheric reanalyses (e. g., Kalnay et al.,

1996; Dee et al., 2011; Rienecker et al., 2011) typically have a horizontal resolution of $\sim 100\,\mathrm{km}$ or less, the resolution of operational forecast products has been continuously improving during the last decades. In this study we use horizontal and





vertical winds from European Centre for Medium-range Weather Forecasts (ECMWF) operational analyses and forecasts[1] for the years 2014 and 2015 produced in spectral truncation T1279, which corresponds to a horizontal resolution of about 16 km. Vertically, the data consists of 137 levels reaching from the surface to 0.01 hPa. For usage with MPTRAC, the wind fields have been interpolated horizontally to a longitude-latitude grid with $0.125° \times 0.125°$ resolution and vertically to 114 pressure levels

in the troposphere and stratosphere up to 5 hPa. 12-hourly analyses are combined with short-term forecasts in between to obtain data with a 3-hour time step. Hoffmann et al. (2016) showed that this data set outperforms existing reanalysis data products in terms of transport deviations for simulations of volcanic sulfur dioxide emissions in the upper troposphere and stratosphere.

Example wind fields from the operational data are presented in Figure 1. Horizontal and vertical wind velocities from the ECMWF operational analysis for 1 January 2015, 00:00 UTC are shown for three pressure levels in the stratosphere, in the

UT/LS region, and in the free troposphere. At about 24 km altitude the global wind fields are dominated by a meandering band of high horizontal wind speed at high northern latitudes indicating the wintertime polar vortex, together with weaker tropical easterlies. Wind dynamics in the extratropical summer hemisphere are generally slow. Enhanced horizontal wind speeds at about 12 km altitude are connected with UT/LS jet streams over both hemispheres and are highest for the subtropical jet stream situated at around 30°N with maxima over the western Pacific reaching more than $100 \, \mathrm{m \, s^{-1}}$ locally. In the free troposphere

typical weather patterns from the moving high and low pressure systems over the mid latitudes exhibit the highest horizontal wind speeds, but with stronger spatial variability than in the stratosphere. The vertical wind velocities mostly vary on short spatial scales of several 100 km or less, often associated with atmospheric gravity waves (e. g. Preusse et al., 2009; Hoffmann et al., 2013). In the troposphere, also contiguous areas of high vertical velocities with extension of 1000 km or more occur close to strong pressure systems. Other high vertical wind speeds are connected with the polar vortex and the jet streams.

Strong vertical winds are also observed at the Inter-Tropical Convergence Zone (ITCZ) which is located around 10°N-20°S for January. Note that many of the small-scale features identified here cannot be found in lower resolution data sets such as global meteorological reanalyses.

### 2.3  Numerical methods for trajectory calculations

Lagrangian particle dispersion models calculate the trajectories of individual particles or infinitesimally small air parcels over

time. The trajectory of each air parcel is defined by the kinematic equation of motion,

$$\frac{d\mathbf{x}}{dt} = \mathbf{v}\left(\mathbf{x}(t), t\right). \tag{1}$$

Here $\mathbf{x} = (x, y, z)$ denotes the position and $\mathbf{v} = (u, v, \omega)$ the velocity of the air parcel at time $t$. In MPTRAC the horizontal position $(x, y)$ of the air parcel is defined by longitude and latitude, which requires spherical coordinate transformations to relate it to the horizontal wind $(u, v)$. The vertical coordinate $z$ is related to pressure $p$ by the hydrostatic equation, and the

vertical velocity is given by $\omega = dp/dt$. The wind vector $\mathbf{v}$ at any position $\mathbf{x}$ is obtained by means of a 4-D linear interpolation of the meteorological data, which is a common approach in many LPDMs (Bowman et al., 2013). The analytic solution of the

---

[1]See http://www.ecmwf.int/en/forecasts/datasets (last access: 8 December 2016).





kinematic equation of motion is given by

$$\mathbf{x}(t_1) = \mathbf{x}_0 + \int_{t_0}^{t_1} \mathbf{v}\left(\mathbf{x}(t), t\right) dt, \tag{2}$$

with initial position $\mathbf{x}_0$ at start time $t_0$ and end time $t_1$. In this study the performance of six numerical schemes to solve the kinematic equation of motion is analyzed. All schemes belong to the class of explicit Runge-Kutta methods, for an overview

of these methods see, e. g., Butcher (2008).

The explicit Euler method likely poses the most simple way to solve the kinematic equation of motion. The numerical solution is obtained from Equation (2) by means of a 1st-order Taylor series approximation. Hence, it is also referred to as 'zero acceleration' scheme. The iteration scheme of the explicit Euler method (referred to as the Euler method below) is given by

$$\mathbf{x}_{n+1} = \mathbf{x}_n + \Delta t\, \mathbf{v}\left(\mathbf{x}_n, t_n\right), \tag{3}$$

where $\Delta t = t_{n+1} - t_n$ refers to the time step. The Euler method is a 1st-order Runge-Kutta method, i. e., the local truncation error for each time step is on the order of $\mathcal{O}(\Delta t^2)$, whereas the total accumulated error at any given time is on the order of $\mathcal{O}(\Delta t)$.

MPTRAC currently uses the explicit midpoint method as its default numerical integration scheme,

$$\mathbf{x}_{n+1} = \mathbf{x}_n + \Delta t\, \mathbf{v}\left(\mathbf{x}_n + \frac{\Delta t}{2}\mathbf{v}\left(t_n, x_n\right), t_n + \frac{\Delta t}{2}\right). \tag{4}$$

First the 'mid point' is calculated using an Euler step with half the time step, $\Delta t/2$. The final step is calculated using the wind vector at the mid point of the Euler step. The midpoint method is a 2nd-order Runge-Kutta method. The local error is on the order of $\mathcal{O}(\Delta t^3)$, giving a total accumulated or global error on the order of $\mathcal{O}(\Delta t^2)$. The method is computationally more expensive than the Euler method, but errors generally decrease faster in the limit $\Delta t \to 0$.

The scheme of Petterssen (1940) is popular in many LPDMs (e. g. Stohl, 1998; Bowman et al., 2013). It is defined by

$$\mathbf{x}_{n+1,0} = \mathbf{x}_n + \Delta t\, \mathbf{v}\left(\mathbf{x}_n, t_n\right), \tag{5}$$

$$\mathbf{x}_{n+1,l} = \mathbf{x}_n + \frac{\Delta t}{2}\left(\mathbf{v}\left(x_n, t_n\right) + \mathbf{v}\left(\mathbf{x}_{n+1,l-1}, t_{n+1}\right)\right), \tag{6}$$

with $l$ being an index counting the number of inner iterations carried out as part of each time step. If no inner iterations are performed, the scheme is equivalent to the Euler method. If one inner iteration is carried out, the method is also known as

Heun's method, another type of a 2nd-order explicit Runge-Kutta method. An increasing number of inner iterations can help to improve the accuracy of the solution in situations with rather complex wind fields. If the local wind field is smooth, it results in fewer iterations and less computing time. We applied the Petterssen scheme with up to 7 inner iterations and did not tune the convergence limit for the inner iterations for efficiency, as we were mostly interested in good accuracy of the solutions.





In this study we also evaluated 3rd- and 4th-order explicit Runge-Kutta methods (RK3 and RK4). The 3rd-order method used here is defined by

$$\mathbf{x}_{n+1} = \mathbf{x}_n + \Delta t \left( \frac{1}{6}\mathbf{k}_1 + \frac{4}{6}\mathbf{k}_2 + \frac{1}{6}\mathbf{k}_3 \right), \tag{7}$$

$$\mathbf{k}_1 = \mathbf{v}\left(\mathbf{x}_n, t_n\right), \tag{8}$$

$$\mathbf{k}_2 = \mathbf{v}\left(\mathbf{x}_n + \frac{\Delta t}{2}\mathbf{k}_1, t_n + \frac{\Delta t}{2}\right), \tag{9}$$

$$\mathbf{k}_3 = \mathbf{v}\left(\mathbf{x}_n - \Delta t\,\mathbf{k}_1 + 2\,\Delta t\,\mathbf{k}_2, t_n + \Delta t\right). \tag{10}$$

The classical 4th-order Runge-Kutta method is defined by

$$\mathbf{x}_{n+1} = \mathbf{x}_n + \Delta t \left( \frac{1}{6}\mathbf{k}_1 + \frac{2}{6}\mathbf{k}_2 + \frac{2}{6}\mathbf{k}_3 + \frac{1}{6}\mathbf{k}_4 \right), \tag{11}$$

$$\mathbf{k}_1 = \mathbf{v}\left(\mathbf{x}_n, t_n\right), \tag{12}$$

$$\mathbf{k}_2 = \mathbf{v}\left(\mathbf{x}_n + \frac{\Delta t}{2}\mathbf{k}_1, t_n + \frac{\Delta t}{2}\right), \tag{13}$$

$$\mathbf{k}_3 = \mathbf{v}\left(\mathbf{x}_n + \frac{\Delta t}{2}\mathbf{k}_2, t_n + \frac{\Delta t}{2}\right), \tag{14}$$

$$\mathbf{k}_4 = \mathbf{v}\left(\mathbf{x}_n + \Delta t\,\mathbf{k}_3, t_n + \Delta t\right). \tag{15}$$

For these methods the local truncation error is on the order of $\mathcal{O}(\Delta t^{p+1})$, while the total accumulated error is on the order of $\mathcal{O}(\Delta t^p)$, with $p$ referring to the order of the method. The classical 4th-order Runge-Kutta method is the highest order Runge-Kutta method for which the number of function calls matches its order. It typically provides a good ratio of accuracy and computation time. Any 5th-order method requires at least six function calls, which causes more overhead.

## 2.4 Evaluation of trajectory calculations

A common way to compare sets of test and reference trajectories is to calculate transport deviations (Kuo et al., 1985; Stohl et al., 1995; Stohl, 1998). Transport deviations are calculated by averaging the individual distances of corresponding air parcels from the test and reference data sets at a given time. The reference data set could be the known analytical solution for an idealized test case, it could be based on observations like balloon trajectories, or it could be obtained by using a numerical integration method known to be highly accurate for real wind data. Absolute horizontal and vertical transport deviations at time $t$ are calculated according to

$$\text{AHTD}(t) = \frac{1}{N} \sum_{i=1}^{N} \sqrt{[X_i(t) - x_i(t)]^2 + [Y_i(t) - y_i(t)]^2}, \tag{16}$$

$$\text{AVTD}(t) = \frac{1}{N} \sum_{i=1}^{N} |Z_i(t) - z_i(t)|. \tag{17}$$

with $X_i(t)$, $Y_i(t)$, and $Z_i(t)$ as well as $x_i(t)$, $y_i(t)$, and $z_i(t)$ referring to the air parcel coordinates of the test and reference data set, respectively. Each data set contains $N$ air parcels. Here we calculated the horizontal distances as Cartesian distances of the



air parcel positions projected to the Earth surface. This approach approximates spherical distances with $\geq 99\%$ accuracy for distances up to 3000 km. Vertical distances are calculated based on pressure and the hydrostatic equation. Relative horizontal transport deviations (RHTD) and relative vertical transport deviations (RVTD) are calculated by dividing the absolute transport deviations by the horizontal or vertical path lengths of the trajectories, respectively.

According to the definition, the transport deviations are calculated as mean absolute deviations of the air parcel distances. Although the mean absolute deviation is a rather intuitive approach to measure statistical dispersion, we note that it is not necessarily the most robust measure, as it can be influenced significantly by outliers. Such outliers of rather large individual transport deviations exist in some of our simulations. Strong error growth of individual trajectories can occur once the test and reference trajectories are significantly separated from each other, meaning that the air parcels are located in completely different

wind regimes. To mitigate this issue we decided to report also the median of the absolute and relative transport deviations of the individual air parcels as an additional statistical measure. The median absolute deviation is a much more robust statistical measure. In all cases considered here we found that the median absolute deviation is smaller than the mean absolute deviation. This indicates that the distributions of transport deviations are skewed towards larger outliers. Note that skewed distributions of transport deviations have also been reported in other LPDM intercomparison and validation studies (e. g., Stohl et al., 2001).

## 2.5    Considerations on time steps and error limits

Since our test cases are based on real meteorological data, we obtained the reference data to calculate the transport deviations using the most accurate integration method available to us with a sufficiently short time step. Tests showed that the numerical solutions from the RK4 method converge for time steps of 60 s or less. In particular, comparing simulations with time steps of 120 s and 60 s, the median horizontal deviation is less than 7 km and the median vertical deviation is less than 10 m up to

10 days of simulation time. Alternatively, following Seibert (1993), we may also evaluate the Courant-Friedrichs-Lewy (CFL) criterion, $\Delta t \leq \Delta x / u_{max}$, to establish a time step estimate for the reference simulations. Based on an effective horizontal resolution of $\Delta x \sim 16$ km and a maximum horizontal wind speed of $u_{max} \sim 120\,\mathrm{m\,s^{-1}}$, we find that $\Delta t \leq 130$ s is needed to ensure sufficiently fine sampling of the ECMWF data. Therefore, we selected a time step of 60 s to calculate the reference trajectories.

25       The maximum tolerable error limits for trajectory calculations depend on the individual application of course. However, as a guideline, we here provide physically motivated error limits that are of particular interest regarding LPDM simulations. LPDMs consider both, advection and diffusion, to calculate dispersion. Clearly, the numerical errors of the trajectory calculations, representing the advective part, should be smaller than the particle spread caused by diffusion. Considering a simple model of Gaussian diffusion, the standard deviations of the horizontal and vertical particle distributions are given by $\sigma_x = \sqrt{2D_x t}$ and

$\sigma_z = \sqrt{2D_z t}$, respectively. Typical vertical diffusivity coefficients are $D_z \sim 1\,\mathrm{m^2\,s^{-1}}$ in the free troposphere (Pisso et al., 2009) and $D_z \sim 0.1\,\mathrm{m^2\,s^{-1}}$ in the lower stratosphere (Legras et al., 2003). Assuming a typical scale ratio of horizontal to vertical wind velocity of $\sim 200$ (Pisso et al., 2009), corresponding horizontal diffusivity coefficients are $D_x \sim 40\,000\,\mathrm{m^2\,s^{-1}}$ in the troposphere and $D_x \sim 4000\,\mathrm{m^2\,s^{-1}}$ in the stratosphere. The corresponding horizontal spread after 10 days is $\sigma_x \sim 260$ km in the troposphere and $\sigma_x \sim 85$ km in the stratosphere. The vertical spread is $\sigma_z \sim 1300$ m in the troposphere and $\sigma_z \sim 415$ m in





the stratosphere. However, note that these values should only be considered as a guideline. Local diffusivities may be an order of magnitude smaller or larger than these values, depending on the individual atmospheric conditions.

### 2.6 Experiment configuration

In this study we analyzed the truncation errors of trajectory calculations in 15 domains of the atmosphere, covering rather
distinct conditions in terms of pressure, temperature, and winds. The globe was divided into 5 latitude bands: polar latitudes ($90°$S to $65°$S and $65°$N to $90°$N; $23.9 \times 10^6$ km$^2$ surface area in each hemisphere), mid-latitudes ($65°$S to $20°$S and $20°$N to $65°$N; $143.9 \times 10^6$ km$^2$ surface area in each hemisphere), and tropical latitudes ($20°$S to $20°$N; $174.2 \times 10^6$ km$^2$ total surface area). The selected 3 altitude layers cover the free troposphere (2 to 8 km; 24 ECMWF model levels), the UT/LS region (8 to 16 km; 24 levels), and the lower and mid stratosphere (16 to 32 km; 31 levels). These domains are of major interest regarding
various kinds of transport simulation applications using MPTRAC and other LPDMs. The planetary boundary layer was not considered here, because MPTRAC lacks more sophisticated parametrization schemes for diffusion needed for simulations in this layer. As the atmospheric conditions depend on the season and vary from year to year, we selected 1 January, 1 April, 1 July, and 1 October of the years 2014 and 2015 as start times for the simulations. All simulations cover a time period of 10 days. In each domain 500,000 trajectory seeds were uniformly distributed. Although this is already a large number of trajectory
seeds, it needs to be pointed out that this is undersampling the effective resolution of the ECMWF data by as much as a factor of 4.5 in the polar troposphere up to a factor of 42 in the tropical stratosphere. Nevertheless, initial tests with different numbers of trajectory seeds showed that our results are statistically significant. In all domains we tested time steps of 60, 120, 240, 480, 900, 1800, and 3600 s for each of the six integration schemes. In total more than 5000 individual transport simulations were performed.

In this study we defined the atmospheric domains by means of fixed latitude and altitude boundaries. This is arguably a rather simple approach compared to physically motivated separation criteria based on equivalent latitudes or the dynamical tropopause. However, the simple approach may still reflect how the model is initialized and used in different applications in practice. An important consequence of the simple approach is that part of the air parcels left their initial domain during the course of simulation. Table 1 provides the fraction of air parcels that remain in their initial domain after 5 and 10 days
simulation time. In the stratosphere we found fractions of $48 - 88\%$ after 5 days and $36 - 78\%$ after 10 days in the different latitude bands. In the UT/LS region the fractions are lower, i. e., $32 - 55\%$ after 5 days and $14 - 40\%$ after 10 days. In the troposphere the fraction is even lower, i. e., $32 - 48\%$ after 5 days and $10 - 24\%$ after 10 days. The lowest fractions are found for the polar latitudes for all altitude layers, being the smallest regions. The horizontal wind maps shown in Fig. 1 suggest that planetary wave activity and meandering of the westerly jets between mid and high latitudes are responsible for the low
fractions at polar latitudes. We also found that the fractions decrease from the stratosphere to the troposphere. This may be attributed to stronger turbulent transport associated with deep convection and eddy diffusivity in the troposphere. Although a substantial fraction of air parcels may leave their initial region during the simulations, we decided to not filter and exclude those trajectories in our analyses. The trajectories that leave the domains are more likely related to larger winds and vertical





velocities. Excluding those trajectories would cause a strong bias towards short trajectories, representing only the lower winds and velocities in the statistical analysis.

## 3 Results

### 3.1 Case studies of trajectory calculations

In this section we present two case studies that illustrate some of the common features related to trajectory calculations using different numerical integration schemes. Figure 2 shows maps of trajectories that were calculated using the six numerical schemes introduced in Sect. 2.3 with a time step of 120 s. Figure 3 provides the corresponding absolute transport deviations with respect to the reference calculations (RK4 method with 60 s time step). Both case studies show trajectories that were launched on 1 January 2014 at about 10 km altitude. In the example for the northern hemisphere the trajectories calculated

using the different schemes agree well (AHTD $\leq$ 200 km and AVTD $\leq$ 600 m) for the first six days. After this point the Euler solution shows rapidly growing errors, with an AHTD up to 3900 km and an AVTD up to 4800 m after 8 days. The Petterssen scheme and Heun's method yield AHTDs $\leq$ 200 km and AVTDs $\leq$ 800 m for about 8 days, before they diverge from the reference calculation. The midpoint and RK3 method provide AHTDs $\leq$ 200 km and AVTDs $\leq$ 800 m until the end of the simulation (after 10 days). The example for the southern hemisphere illustrates that the onset of rapid error growth may

occur much earlier in time. Here an AHTD of 200 km and an AVTD of 800 m is already exceeded after 3 days by the Euler solution and after 4 to 6 days by the solutions from Heun's method and Petterssen's scheme. However, although error growth starts earlier, in the southern hemisphere example the maximum AHTD remains below 2200 km and the AVTD below 2200 m, which is factor of 2 lower compared with the northern hemisphere example. Relative transport deviations between the examples are more similar, as the horizontal trajectory length is about 36,400 km in the northern hemisphere, but only 14,400 km in the

southern hemisphere.

A common feature of the trajectory calculations we found in the case studies and also in many other situations is that the numerical integration schemes yield solutions that typically agree well up to a specific point in time before rapid error growth begins. Errors grow slowly in the beginning, but at some point, e. g., if there is strong wind shear locally, the trajectories may begin to diverge significantly. Shorter time steps or high-order integration schemes are needed to properly cope with such

situations. The case studies also show that transport deviations do not necessarily grow monotonically over time. Trajectories may first diverge from and then reapproach the reference data. Individual local wind fields can bring trajectories back together by chance. The case studies also seem to suggest that vertical errors start to grow earlier than horizontal errors. Furthermore, we note that the Petterssen scheme mostly provides smaller errors than Heun's method. This was expected, because the Petterssen scheme provides iterative refinements compared with Heun's method. In both case studies the midpoint method performs better

than the other 2nd-order methods. However, this is not valid in general, we also found counter-examples with the midpoint method performing worse than the other 2nd-order methods. Both examples generally exhibit large variability of the errors. This indicates that transport deviations need to be calculated for large numbers of air parcels to obtain statistically meaningful results.



## 3.2 Growth rates of truncation errors

In this section we discuss the temporal growth rates of the truncation errors of the trajectory calculations from a more general point of view. Although the magnitude of the truncation errors varies largely between the schemes and with the time step used for numerical integration, we found that the transport deviations typically grow rather monotonically over time, if large numbers of particles are considered. Hence, we decided to present here the truncation errors using a fixed time step of $120\,\mathrm{s}$ for the numerical integration as a representative example. As the magnitude of the truncation errors varies largely between the troposphere and stratosphere, we present the analysis for both regions separately. The results for the UT/LS region are not shown, as they just fall in between. We calculated combined transport deviations considering all the seasons and all the latitude bands in the given altitude range. A more detailed analysis of the truncation errors in individual latitude bands and for different seasons will follow in Sect. 3.3. The influence of the choice of the time step on the accuracy and performance of the trajectory calculations will be discussed in Sect. 3.4.

Figure 4 shows the AHTDs and AVTDs of the trajectory calculations for the troposphere and stratosphere as obtained by the different numerical schemes. A common feature is the clustering of the results into three groups, which we attribute to the numerical order of the integration schemes. The largest truncation errors are produced by the Euler method, which is a 1st-order scheme. After 10 days simulation time we found absolute (relative) horizontal transport deviations of $1450\,\mathrm{km}$ (14.6%) in the troposphere and $170\,\mathrm{km}$ (1.4%) in the stratosphere as well as vertical transport deviations of $1150\,\mathrm{m}$ (13.3%) in the troposphere and $194\,\mathrm{m}$ (3.5%) in the stratosphere. The truncation errors of the 2nd-order methods (midpoint, Heun, and Petterssen scheme) are typically $1-2$ orders of magnitude smaller compared to the Euler method. For the midpoint method we found horizontal transport deviations of up to $320\,\mathrm{km}$ (3.4%) in the troposphere and $11\,\mathrm{km}$ (0.086%) in the stratosphere as well as vertical transport deviations of up to $361\,\mathrm{m}$ (3.9%) in the troposphere and $14\,\mathrm{m}$ (0.18%) in the stratosphere. The RK3 and RK4 methods cluster in the third group, with truncation errors being another factor $2-4$ lower than for the 2nd-order schemes. For the RK3 method we found horizontal transport deviations of up to $228\,\mathrm{km}$ (2.5%) in the troposphere and $6.7\,\mathrm{km}$ (0.048%) in the stratosphere as well as vertical transport deviations of up to $272\,\mathrm{m}$ (2.9%) in the troposphere and $8\,\mathrm{m}$ (0.099%) in the stratosphere. We attribute the fact that there are nearly no differences between the RK3 and RK4 method to the use of the 4-D linear interpolation scheme for the meteorological data. A high-order numerical integration schemes is not expected to provide any large benefits in combination with a low-order interpolation scheme.

From the data presented in Fig. 4 we can also estimate the temporal growth rates of the truncation errors as well as the leading polynomial order of the error growth. We found that error growth typically starts off linear, i. e., with a polynomial order close to one, but gets non-linear already after $1-2$ days, with the polynomial order getting significantly larger than 1. For the troposphere we found a maximum polynomial order of $\sim 3$ after 5 days for the AHTDs and of an order $\sim 2$ after 4 days for the AVTDs for the Euler method. The higher order methods show non-linearity at even higher levels, with a maximum polynomial order of $\sim 5$ after 8 days for the AHTDs and of $\sim 4$ after 6 days for the AVTDs for the RK3 and RK4 method. The 2nd-order methods are in between. Due to the non-linear error growth, the growth rates of the truncation errors also increase rapidly over time until they reach their maxima after 10 days. For the Euler method we found horizontal growth rates of

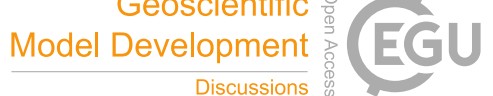



up to 334 km day$^{-1}$ (2.1 percentage points per day) in the troposphere and 43 km day$^{-1}$ (0.26 pp day$^{-1}$) in the stratosphere as well as vertical growth rates of up to 181 m day$^{-1}$ (1.0 pp day$^{-1}$) in the troposphere and 35 m day$^{-1}$ (0.41 pp day$^{-1}$) in the stratosphere. For the midpoint method (representing the 2nd-order methods) we found horizontal growth rates of up to 115 km day$^{-1}$ (0.93 pp day$^{-1}$) in the troposphere and 3.2 km day$^{-1}$ (0.024 pp day$^{-1}$) in the stratosphere as well as vertical

growth rates of 105 m day$^{-1}$ (0.89 pp day$^{-1}$) in the troposphere and 3.3 m day$^{-1}$ (0.042 pp day$^{-1}$) in the stratosphere. For the RK3 method we found horizontal growth rates of up to 87 km day$^{-1}$ (0.73 pp day$^{-1}$) in the troposphere and 1.9 km day$^{-1}$ (0.013 pp day$^{-1}$) in the stratosphere as well as vertical growth rates of 86 m day$^{-1}$ (0.74 pp day$^{-1}$) in the troposphere and 1.9 m day$^{-1}$ (0.024 pp day$^{-1}$) in the stratosphere.

### 3.3 Regional and seasonal truncation errors

For a more detailed analysis of regional and seasonal truncation errors we focus on the errors after 10 days simulation time for simulations using the 3rd-order Runge Kutta method with a single time step of 120 s. This is considered to be a representative example as other schemes and time steps show similar regional and seasonal variations. We calculated individual transport deviations for all 15 altitude-latitude domains and for simulations starting at the beginning of January, April, July, and October 2014 and 2015, respectively. The results are shown in Figs. 5 and 6.

Our simulations show that horizontal errors increase from typically 20 km in the stratosphere to 100 km in the UT/LS region and about 200 km in the troposphere. The corresponding maximum AHTDs are 116 km, 177 km, and 470 km, respectively. The corresponding relative errors increase from 0.0 to 0.4 % in the stratosphere, to around 0.1 to 1.0 % in the UT/LS region, and 1.0 to 4.0 % in the troposphere. As shown in Fig. 4, the truncation errors in the stratosphere comply on average with the error limit defined in Sect. 2.5, while the error limit in the troposphere is reached after about 10 days. However, as can be seen

from Figs. 5 and 6, the truncation errors can vary considerably seasonally and inter-annually as well as between the domains, causing maximum errors of specific domains to exceed the defined limit. In the stratosphere, generally the horizontal transport deviations are smaller than 40 km, far below the error limit. An exception is found for the NH polar stratosphere in January 2015 with an AHTD of 116 km. Errors are growing from the stratosphere towards the troposphere. While stratospheric wind fields are rather uniform, the turbulent wind fraction becomes stronger and more frequent at lower altitudes. Wind speeds and

wind directions can vary strongly in turbulent regions. So, even if travel distances in the troposphere may be relatively short, transport deviations typically increase with decreasing altitude.

Truncation errors at all altitude layers vary with latitude. We focus on the horizontal errors in this case, but vertical errors show similar results. The troposphere has its largest errors at northern mid-latitudes with errors between 245 km and 470 km. The tropospheric mid-latitudes were expected to cause the largest errors, because the most complex wind systems occur in this

region due to a larger land surface ratio and more complex orography. The errors in the polar regions are second largest with average errors of around 200 km and peak errors in polar summer of up to 380 km. The tropics and southern mid-latitudes have small errors of less than 200 km and adhere to the error limit in all test cases. The UT/LS region has its largest AHTDs in the northern mid-latitudes with 95 km to 177 km. These errors are caused by complex wind patterns and higher turbulence in the underlying region. The second largest errors occur in the tropics with about 75 km on average followed by the north pole and





southern mid-latitudes with about 50 km on average. The south pole has the smallest errors in this altitude layer with about 30 km on average. Errors in the stratosphere are generally small, with some larger average errors in the tropics. All test cases show errors between 2 km and 25 km, which is mostly much larger than in the other stratospheric regions. The relative high errors in the tropics are probably caused by a stronger turbulence in that region. The lower bound of the stratospheric region of

our test cases is 16 km, since the tropopause reaches an average altitude of 16 km near the ITCZ, turbulent movements due to deep convection can occur more frequently in the lower stratosphere above the tropics.

The variation of the horizontal integration errors also exhibits some seasonal dependencies. This is most prominent for the northern mid latitudes, where maximum errors in all cases occur in January. During northern hemisphere wintertime land-sea temperature differences as well as the temperature gradient between the North pole and the equator are largest, which allows

for more intense and complex dynamic patterns to occur than in summer. Our test cases for the southern hemisphere and for the arctic region do not show a seasonal behavior as clearly as one could expect. We need to stress that each simulation lasts only 10 days, which is a relatively short time interval to analyze seasonal effects. Fast temporal variations and changes in medium-range weather patterns can blur out the impact of seasons that is observed here. The small error differences between polar summer and winter additionally can be attributed to the small fraction of parcels that stay in that region. Only 13% of

the parcels that are represented by the statistic remained in the polar regions after 10 days of simulation, which weakens our statistics.

As a rough indication for inter-annual variability, we simulated the same domains and periods for 2014 and 2015. Most of our simulations for the corresponding months differ by less than 20 %, only deviations of a few individual months differ more strongly but in a similar range than the seasonal variations. Most striking differences occur in January in the stratosphere of

the northern polar region. The simulation of 2014 shows small errors of 4 km, while the simulation of 2015 reaches an error of up to 116 km and exceeds the stratospheric error limit. This particular behavior (which is also present in Fig. 6 with an AVTD of 132 m) may be related to a specific meteorological situation during the winter 2014/2015, where a sudden stratospheric warming event occurred during the first days of January 2015 and temporarily caused nearly a split of the arctic vortex in the lower stratosphere (Manney et al., 2015). Significant disturbances of the wind field during this event may be a reason why

trajectory calculations exhibit larger errors.

Vertical and horizontal errors behave very similar, extrema are found in the same domains. Vertical transport deviations are about 800 – 1000 times smaller than the horizontal transport deviations. The errors in the stratosphere are usually very small and below 10 m. Typical errors in the UT/LS region and in the troposphere are about 100 m and 250 m, respectively. Corresponding maximum errors are 130 m in the stratosphere, 168 m in the UT/LS region, and 470 m in the troposphere. The

vertical error limits of 415 m in the stratosphere and 1300 m in the troposphere are easily adhered to. Relative vertical errors range 0.0 – 0.9 % in the stratosphere, 0.2 – 1.6 % in the UT/LS region, and 1.2 – 4.4 % in the troposphere.

We also calculated the horizontal and vertical median errors for the regions. In general, horizontal and vertical median errors are much smaller than the mean errors. Small median deviations shows that most trajectories follow closely to the reference. Those parcels that part from the reference usually diverge strongly, which leads to a high average deviation. The median error

gets somewhat larger in the troposphere, where particle paths are more likely being affected by atmospheric turbulence.





To summarize, the relative errors of $2-4\,\%$ in the troposphere show that this layer is more difficult to solve and that relatively large uncertainties remain even if the absolute error limit is adhered to. The stratospheric relative errors of of about $1\,\%$ are less critical for the integration method. The large difference of the truncation errors of the altitude regions suggests that lower order integration schemes or larger time steps could be used in the stratosphere to save computation time without causing significant errors. Tropospheric northern mid-latitudes are most challenging areas for numerical integration.

### 3.4 Computational efficiency

In this section we focus on the computational efficiency of the numerical integration schemes, which is assessed in terms of the trade-off between computational accuracy of and the computational time required for the trajectory calculations. As the computational efficiency depends, to some extent, on the problem size and the computer architecture that is applied, we will discuss the scalability of the application first. Our scalability tests were performed on the Jülich Research on Exascale Cluster Architectures (JURECA) supercomputer (Krause and Thörnig, 2016). JURECA is equipped with two Intel Xeon E5-2680 v3 Haswell central processing units (CPUs) per compute node. Each node is equipped with $2 \times 12 = 24$ physical compute cores, operating at 2.5 GHz clock-speed. The CPUs support 2-way simultaneous multithreading (SMT), i. e., each node provides up to 48 logical cores. A runtime improvement of up to 50% can be expected due to the SMT feature.[2]

As an example, Fig. 7 shows results of scaling tests using the midpoint scheme with a time step of $120\,\text{s}$ for different numbers of particles and OpenMP threads. For large numbers of particles (on the order of $10^4$ to $10^7$) we found that the CPU time scales linearly with the number of particles. The computation per time step and particle requires between $0.31 \times 10^{-6}$ and $9.0 \times 10^{-6}$ s computing time, depending on the number of the OpenMP threads. For small numbers of particles (on the order of 1 to $10^4$) the computing time is limited by an offset of $6.3 \times 10^{-5}$ to $4.3 \times 10^{-3}$ s, which is due to the overhead of the OpenMP parallelization. Figure 7 also shows the speedup of the OpenMP parallelization for growing numbers of threads. We found that the trajectory code provides good to excellent parallel efficiency for large numbers of particles. The computational efficiency is about 83% for up to 24 physical threads and for $10^5$ to $10^6$ particles. It is also found that the code provides additional speedup if the simultaneous multithreading capabilities of the compute nodes are used, in particular for very large numbers of particles (on the order of $10^6$ to $10^7$). For smaller number of particles ($10^4$ or less) the speedup is limited due to the overhead of the OpenMP parallelization and by the limited work load of the problem itself.

As a measure of computational efficiency, Fig. 8 illustrates the trade-off between computational accuracy, in terms of the AHTD, and computational time. In particular, Fig. 8 illustrates how this trade-off depends on the selection of the time step for the different integration schemes. Results are shown separately for the troposphere and stratosphere, as we already discussed in Sects. 3.2 and 3.3 the troposphere is much more challenging for the integration methods than the stratosphere. From this analysis we find that despite being the fastest method, the Euler method usually has the lowest computational efficiency because of its low accuracy. The 2nd-order methods as well as the RK3 and RK4 methods yield much smaller truncation errors, in particular for short time steps. Among the 2nd-order methods the Petterssen scheme has the lowest computational efficiency, which is due to the fact that we tuned the convergence criteria for this method for accuracy rather than speed.

---

[2]See http://www.fz-juelich.de/ias/jsc/EN/Expertise/Supercomputers/JURECA/UserInfo/SMT.html (last access: 12 December 2016).





The best efficiency, i. e., the best accuracy at the lowest computational costs, is mostly obtained with the midpoint and RK3 methods. The RK4 method does not provide any benefits in combination with the low-order 4-D linear interpolation scheme for the meteorological data. In fact, the RK4 method is slightly less efficient than the RK3 method due to the higher numerical costs.

Figure 8 also allows us to more accurately establish the individual optimal time steps of the integration methods with respect to the error limits defined in Sect. 2.5. This approach is similar to the well-known discrepancy principle (Engl et al., 1996), where the time step is considered as a tuning factor so that the truncation errors of the methods match an a priori known error bound. To provide estimates for all methods, we use linear extra- and interpolation to determine the largest time step that just fulfills the error limit. For the troposphere we derived time steps of about 100 s for the Petterssen scheme, Heun's scheme, and
the midpoint method, and about 170 s for the RK3 and RK4 methods. For the stratosphere we found time steps of about 800 s for the Petterssen scheme, Heun's scheme, and the midpoint method, and time steps of about 1100 s for the RK3 and RK4 methods.

## 4   Summary and conclusions

In this study we characterized the regional and seasonal truncation errors of trajectory calculations in the free troposphere, in the
UT/LS region, and in the stratosphere. Transport simulations were conducted with the LPDM MPTRAC, driven by wind fields from ECMWF operational analyses and forecasts in 2014 and 2015, with an effective horizontal resolution of about 16 km. We analyzed the computational performance of the simulations in terms of accuracy and CPU-time costs of six explicit integration schemes that belong to the Runge Kutta family. The truncation errors of the schemes were found to cluster into three groups that are related to the order of the method: (i) the 1st-order Euler method, (ii) the 2nd-order methods (midpoint method, Heun's
method, and Petterssen's scheme), (iii) the higher order methods, which are the common RK3 and RK4 methods. Different methods within each group provide similar accuracy in terms of error growth rates and transport deviations.

Based on more than 5000 individual transport simulations, we further analyzed horizontal and vertical transport deviations in relation to altitude, latitude, as well as seasonal and year-to-year variability. We found that tropospheric simulations require more accurate integration methods or significantly shorter time steps to keep errors within physically motivated error limits
than simulations for the stratosphere. We attribute this to larger small-scale variations caused by atmospheric turbulence and mixing in the troposphere. Truncation errors also depend on the latitude band, with the northern mid-latitudes having the largest errors in each altitude layer. Seasonal and inter-annual error variations are clearly visible from our simulations, but in some cases the number of samples still seems to be too small to deduce robust statistics. One example are large errors that are associated with a sudden stratospheric warming in the northern stratosphere in January 2015, which suggests that part of the
truncation errors is due to situation-dependent factors. However, a robust feature seams to be a northern mid-latitude winter maximum in the troposphere and stratosphere, existent in both years, 2014 and 2015.

All integration methods discussed here are in principle suited and have been used for LPDM simulations. To decide which method is most efficient on state-of-the-art high performance computing systems, we analyzed the trade-off between compu-





tational accuracy and computational time. This trade-off is largely controlled by the time step used for numerical integration. The Euler method requires very short time steps to achieve reasonably accurate results and is therefore not considered to be an efficient method. Heun's method and the iterative Petterssen scheme are more accurate at the same computational costs. The midpoint method and the RK3 method usually lead to the most efficient simulations, i. e., these methods provide the most

accurate results at the lowest computational costs. Note that the RK4 method is slightly more expensive than the RK3 method if it is applied together with a low-order linear interpolation scheme for the meteorological data.

The study of Seibert (1993) addressed the choice of the numerical integration scheme and choice of the time step based on idealized test cases and for realistic wind fields. To achieve truncation errors that are smaller than overall trajectory uncertainty, they found that the time step should fulfill the CFL criterion as a necessary condition for convergence. This new study used

meteorological data with a very fine spatial resolution as used in current global weather forecast models, which requires adjustment of the time step. Time steps of 10 minutes to 1 hour as used in former trajectory studies (e. g. Seibert, 1993; Stohl et al., 1998, 2001; Harris et al., 2005) are far beyond yielding convergence with high-resolution meteorological data. Given a effective horizontal resolution of 16 km and applying the CFL criterion, the time step needs to be shorter than about 130 s. From our simulations we found that time step of 100 s (midpoint method) and 170 s (RK3 method) provide accurate results for

the troposphere. Purely stratospheric applications can be solved with time steps of 800 s (midpoint method) and 1100 s (RK3 method) because of lower truncation errors in this altitude layer.

In this study we considered a range of popular and well-established integration schemes for trajectory calculations in LPDMs. However, the large variability of regional and seasonal truncation errors found here suggests that applications may benefit from more advanced numerical techniques. Adaptive quadrature could be an interesting topic for future research.

**5   Code and data availability**

Operational analyses and forecasts are distributed by the European Centre for Medium-Range Weather Forecasts (ECMWF), see http://www.ecmwf.int/en/forecasts/datasets (last access: 21 December 2016) for further details. The code of the Massive-Parallel Trajectory Calculations (MPTRAC) model is available under the terms and conditions of the GNU General Public License, Version 3 from the repository at https://github.com/slcs-jsc/mptrac (last access: 21 December 2016).

*Acknowledgements.* The authors acknowledge the Jülich Supercomputing Centre (JSC) for providing computing time on the supercomputer JURECA. YH also acknowledges support from the "100 Talents Program" of Sun Yat-sen University, Special Program for Applied Research on Super Computation of the NSFC-Guangdong Joint Fund (the second phase), and the National Supercomputer Center in Guangzhou. We thank Xue Wu for helpful comments on an earlier draft of this manuscript.



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



**Table 1.** Fractions of Air Parcels Remaining in Initial Regions during Course of Simulations

| | SH Polar Lat. | SH Mid Lat. | Tropical Lat. | NH Mid. Lat. | NH Polar Lat. |
| --- | --- | --- | --- | --- | --- |
| | $(90°S-65°S)$ | $(65°S-20°S)$ | $(20°S-20°N)$ | $(20°N-65°N)$ | $(65°N-90°N)$ |
| After 5 Days Simulation Time: | | | | | |
| Stratosphere ($16-32$ km) | 66% | 86% | 88% | 78% | 48% |
| UT/LS Region ($8-16$ km) | 42% | 55% | 44% | 53% | 32% |
| Troposphere ($2-8$ km) | 32% | 46% | 48% | 44% | 32% |
| After 10 Days Simulation Time: | | | | | |
| Stratosphere ($16-32$ km) | 54% | 77% | 78% | 67% | 36% |
| UT/LS Region ($8-16$ km) | 25% | 40% | 19% | 36% | 14% |
| Troposphere ($2-8$ km) | 13% | 21% | 24% | 20% | 10% |





**Figure 1.** ECMWF operational analysis horizontal wind speed (left) and vertical velocity (right) at about 24 km (top), 12 km (middle), and 5 km (bottom) altitude on 1 January 2015, 00:00 UTC. Black lines indicate the latitude bands considered in our analysis.





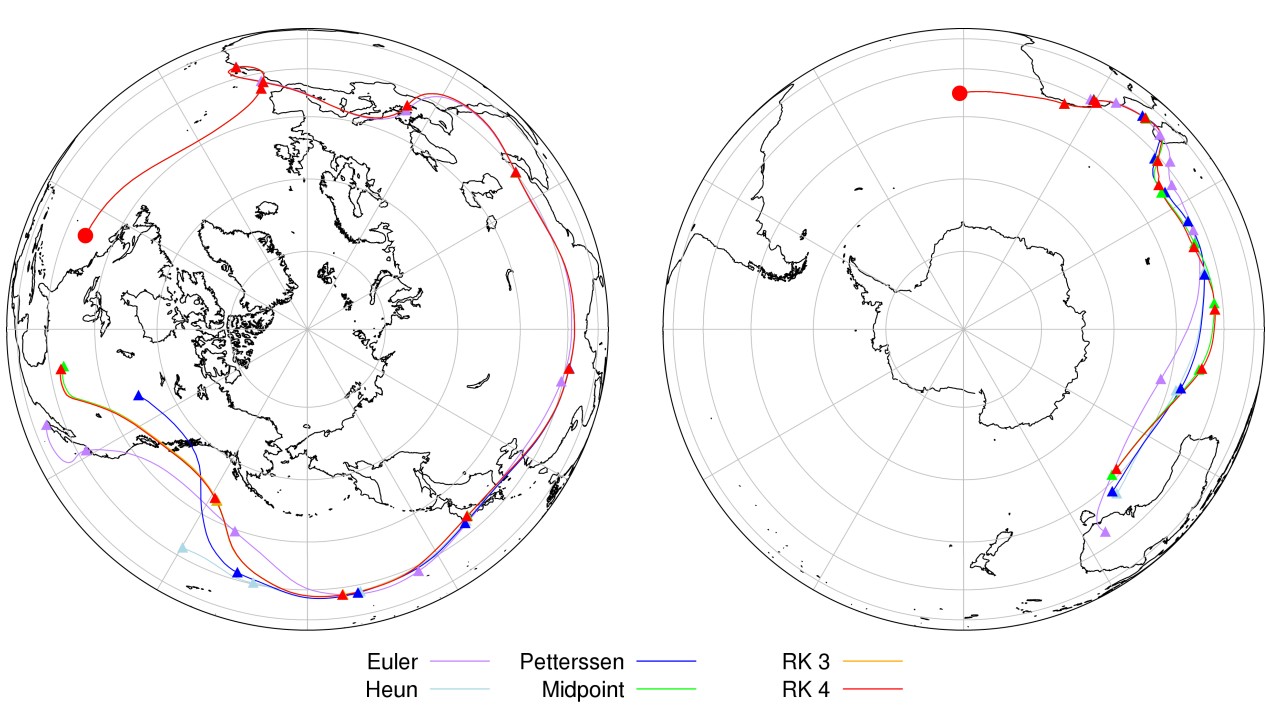

**Figure 2.** Examples of trajectory calculations using different numerical integration schemes. Circles mark the start positions of the trajectories. Trajectories were launched at an altitude of 10.8 km (left) and 9.7 km (right). The start time is 1 January 2014, 00:00 UTC for both. Triangles mark trajectory positions at 00:00 UTC on each day.





**Figure 3.** Absolute horizontal (top) and vertical (bottom) transport deviations for the case studies for the northern hemisphere (left) and southern hemisphere (right) presented in Fig. 2. Please note different ranges of y-axes. Results of the RK3 and RK4 method are close to zero in most cases.







**Figure 4.** Absolute horizontal (left) and vertical (right) transport deviations of trajectory calculations for the stratosphere (top) and tropo-sphere (bottom). The trajectory calculations are based on different numerical schemes, but use the same time step ($\Delta t = 120\,\mathrm{s}$). Grey lines show error limits based on particle diffusion.





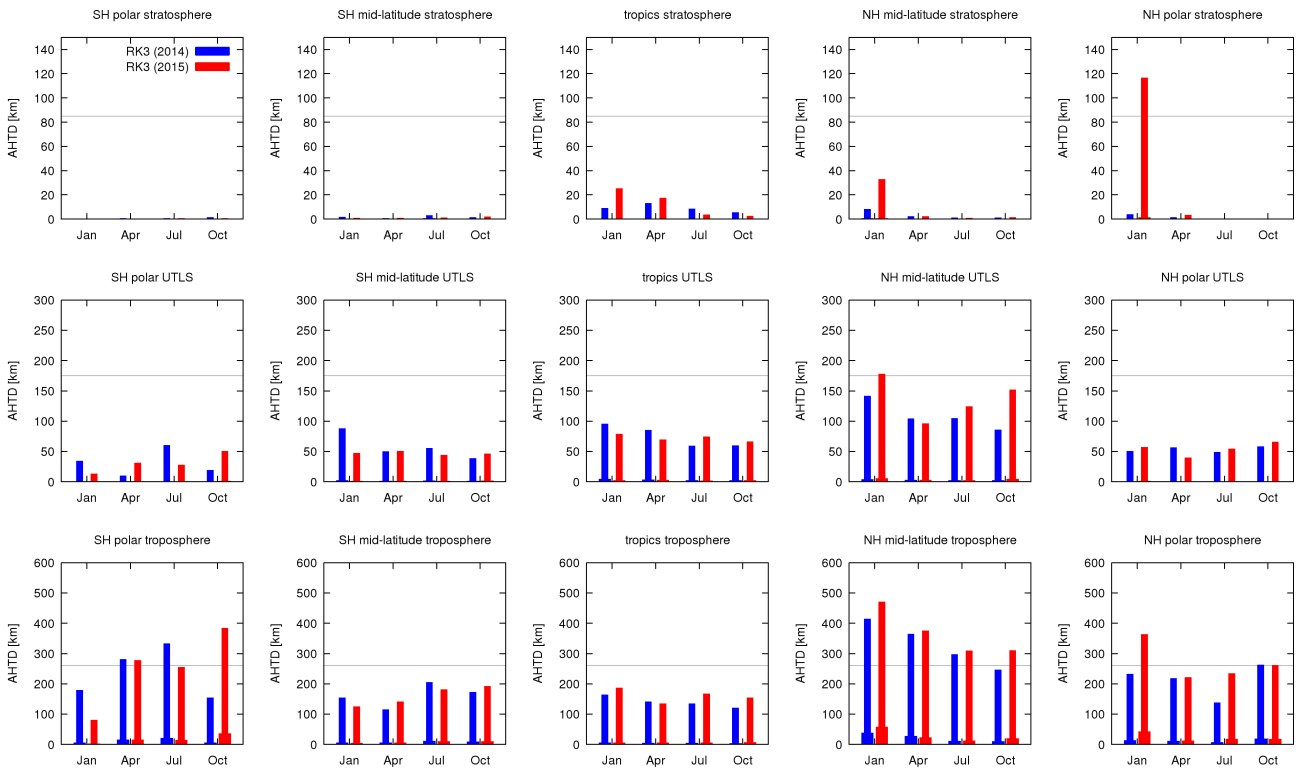

**Figure 5.** Mean (thin bars) and median (thick bars) horizontal transport deviations after 10 days simulation time in different domains for the RK3 method and 120 s time step. Gray lines show error limits based on diffusion.





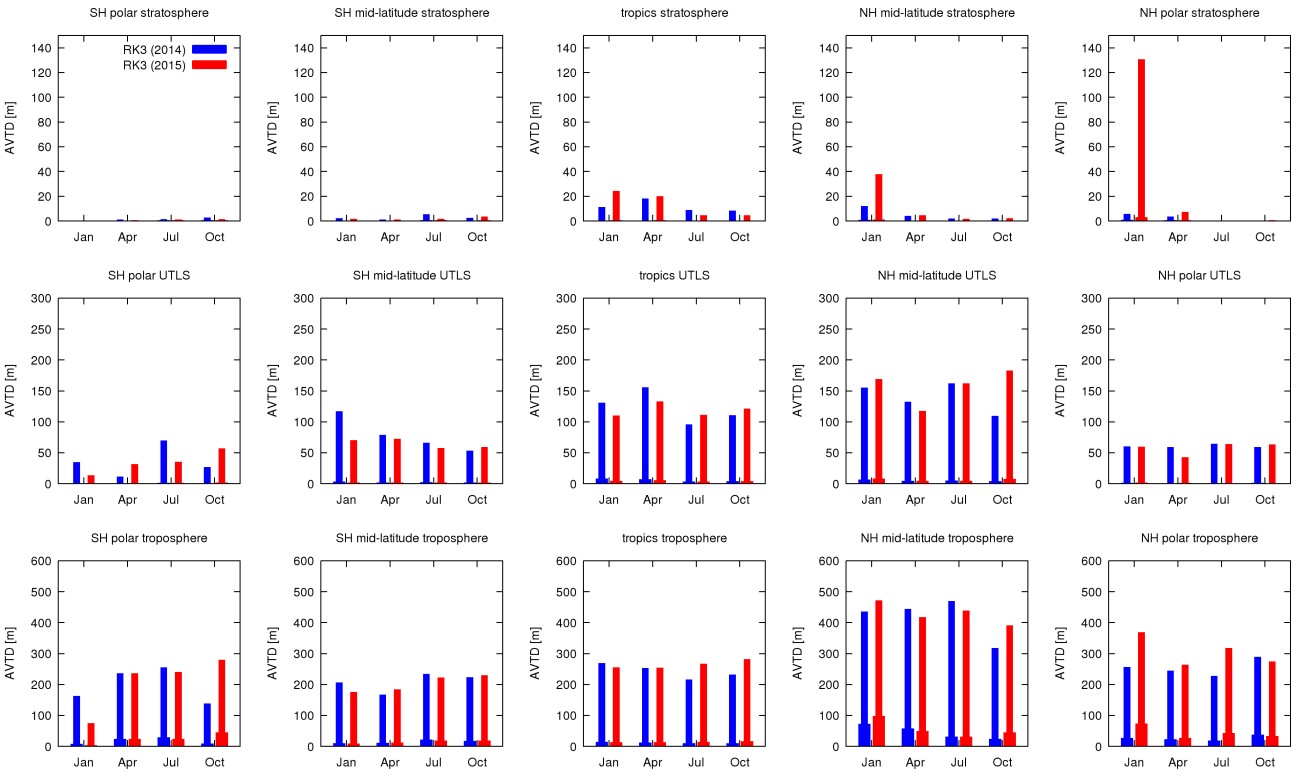

**Figure 6.** Same as Fig. 5, but for vertical transport deviations. Error limits based on diffusion are about 1300 m for the troposphere and 415 m for the stratosphere, which is beyond the AVTD ranges shown here.





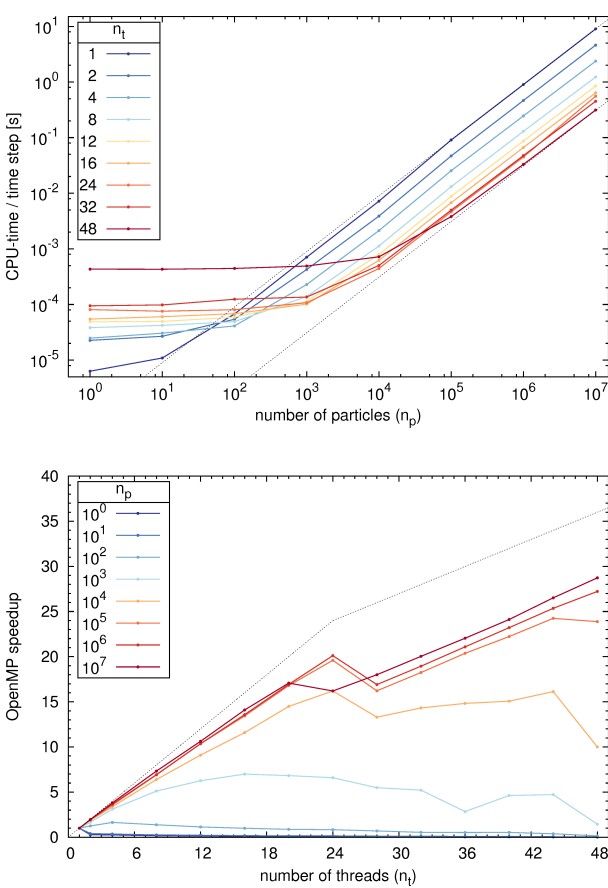

**Figure 7.** Scaling behaviour in terms of CPU-time (top) and speedup of the code (bottom) used to calculate trajectories with the mid-point method and a time step of 120 s for different numbers of particles ($n_p$) and OpenMP threads ($n_t$). Colored curves refer to different numbers of OpenMP threads (top) or different total numbers of particles (bottom). Dotted lines show ideal scaling behavior.



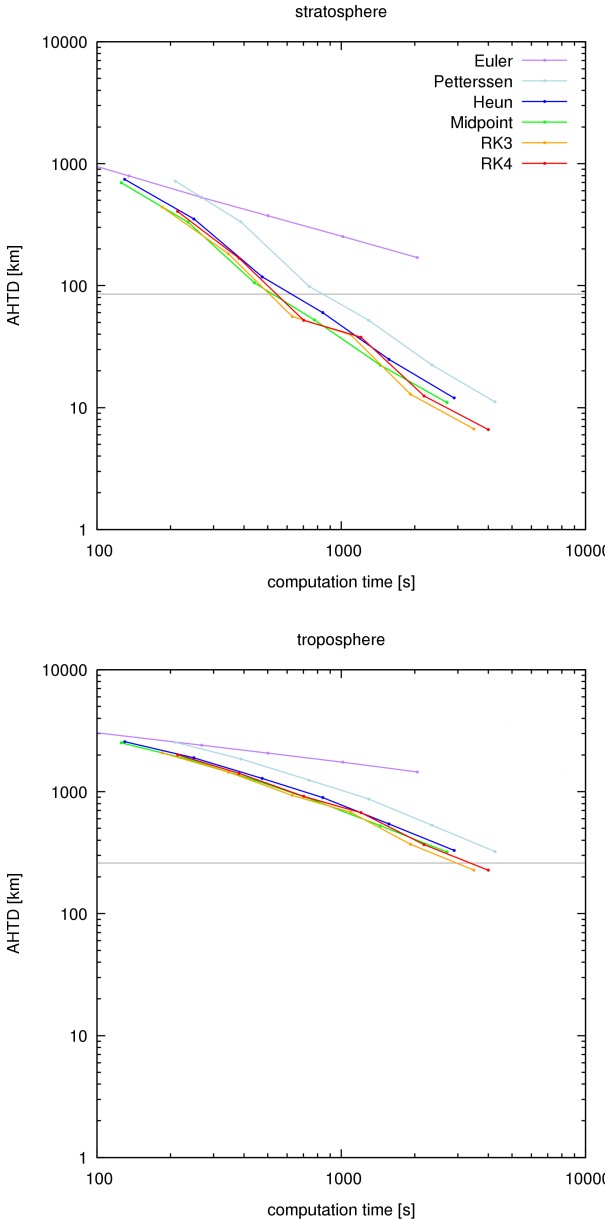

**Figure 8.** Trade-off between computational accuracy and total CPU-time requirements of the trajectory calculations. Colored curves refer to different integration schemes. Dots along the curves indicate time steps of 3600, 1800, 900, 480, 240, and 120 s (from left to right).