# Peer review of "Trajectory errors of different numerical integration schemes diagnosed with the MPTRAC advection module driven by ECMWF operational analyses"

_Geoscientific Model Development, 2016_

## Short Comment (SC1) · 24 Feb 2017

Dear authors,

in my role as Executive editor of GMD, I would like to bring to your attention our Editorial version 1.1:

http://www.geosci-model-dev.net/8/3487/2015/gmd-8-3487-2015.html

This highlights some requirements of papers published in GMD, which is also available on the GMD website in the 'Manuscript Types' section:

http://www.geoscientific-model-development.net/submission/manuscript_types.html

[Figure]

In particular, please note that for your paper, the following requirements have not been met in the Discussions paper:

- "The main paper must give the model name and version number (or other unique identifier) in the title."

- "If the model development relates to a single model then the model name and the version number must be included in the title of the paper. If the main intention of an article is to make a general (i.e. model independent) statement about the usefulness of a new development, but the usefulness is shown with the help of one specific model, the model name and version number must be stated in the title. The title could have a form such as, "Title outlining amazing generic advance: a case study with Model XXX (version Y)"."

So please add the model name and/or ist acronym (MPTRAC) and its respective version number in the title of your article in your revised submission to GMD.

Yours,

Astrid Kerkweg

---

## Referee Comment (RC1) · P. Seibert (Referee) · 20 Mar 2017

**1 General**

This paper investigates the performance of a number of numerical schemes to integrate the trajectory equation. This is done using the LPDM MPTRAC. As the code is prepared for parallel computing, the performance is investigated also as a function of the number of threads (for one of the schemes only). For the tests, 10-day simulations were carried out using ECMWF data with 16 km grid spacing, and results are presented for different regions of the globe, layers of the atmosphere, and seasons.

This study is a useful addition to the previous investigations, as it also tests higher-order methods rarely used in atmospheric transport modelling, and as parallel performance is included. I recommend publishing it after consideration of the following remarks. I think that the authors have some choices with respect to doing additional calculations and/or evaluations, and I hope that they would be able to consider my respective suggestions, as the value of this work could be siginificantly increased in that way.

**2 Major remarks**

1. In my opinion, there is one aspect in the setting of the numerical experiments which is not ideal. The regular LPDM code has been used, that is, including a stochastic wind component to represent turbulence. Existing similar studies have been carried out with simple trajectory models. It is not very clear what the consequence of adding stochastic wind components is for the deviations between the schemes tested. The authors propose turbulence as explanation for several of the observed variations in accuracy, but this remains hypothetic. I would strongly recommend to repeat at least a subset of the simulations with all kinds of stochastic influences (turbulence, mesoscale fluctuations, convection if it exists in the model) switched off, present and discuss these results as well.

2. Another open question is whether RK4 with 60 s time step is a suitable reference method. If one extrapolates the RK3 or RK4 curves in Fig. 8 (bottom), one would arrive at an AHTD value of about 100 km at 60 s (probably against a hypothetical perfect simulation). The time step has to be reduced until a further reduction does not reduce AHTD significantly in order to establish a reference simulation. (I see that Hoffmann et al. (2016) claim that convergence already was reached at 120 s, but this is in obvious contradiction with the results reported here.) This might change the apparent relative benefits of higher-order methods.

[Figure]

3. As the authors rightly point out, higher-order methods are unlikely to bring much gain if we use linear interpolation. This points to another option for a potentially optimal trajectory calculation, at least as a reference method: Linear interpolation should allow to solve the trajectory equation analytically within a grid cell and between two times of wind field availability (cf. Seibert, 1993). Admittedly, the need to bound each calculation step at grid-cell borders has a potential to make this method a bit cumbersome and computationally probably less efficient.

4. Another methodological issue is the questions on which transport times the final evaluation of schemes should be based. Even though not explicitly mentioned, Fig. 8 seems to be made with results after 10 days. I don't think this is the most appropriate choice. As discussed in Section 3.2, there is a strongly non-linear growth of the deviations with time. This growth has nothing to do with numerical errors, it is solely a function of atmospheric flow patterns (diffluent flows or bifurcations). Thus, a longer calculation mainly amplifies initial deviations which are due to the different truncation errors. The longer calculations only mean more calculational efforts, and the true truncation errors are obscured by the increasingly important atmospheric flow influences, probably exaggerating the difference between atmospheric regions or seasons (note also that for example polar-region trajectories mostly leave the polar domain within the 10 days). Please also look at results with much shorter transport times and consider replacing the 10-day results by them.

5. Finally, the results are certainly sensitive to the resolution of the wind field data. Results obtained for the specific case of 16 km / 3 h therefore cannot be generalised. Keeping in mind the conclusions of Stohl et al. (1995), Brioude et al. (2012), and Bowman et al. (2013), 3 h intervals for the wind fields are coarser than what would be desired at this horizontal resolution. As 1 h is provided by ECMWF, I am wondering why it was not used. This also diminishes the value of the results presented here, as most people would want to use the 1-h data if

they go to the highest horizontal resolution. There would be a number of ways to produce more general results, such as trying out different resolutions or to parameterise the recommended time step by flow field properties such as (local) spatial and/or temporal derivatives at different orders.

**3  Specific and Minor Remarks**

1. The title could be rephrased for example as "Truncation errors of trajectory calculations using ECMWF high-resolution data diagnosed with the MPTRAC Lagrangian particle dispersion model"

2. Page 1, line 1: *Abstract*. The abstract could be shortened by removing non-essential background and more concise wording.

3. Page 1, line 4: *kinematic equation of motion* (comes also in other places). I don't feel comfortable with this wording. "Equations of motion" for me would refer to the Euler or Navier-Stokes equations. Why not call this the *trajectoy equation*?

4. Page 2, line 6: *Lagrangian particle dispersion models have proven*. Under this chapeau, next to real LPDMs, LAGRANTO is listed which is a simple trajectory model and not an LPDM. I think it does no harm to enumerate it here, but not under a category that doesn't fit (and there is no reason to focus specifically on LPDMs here, as the truncation error problem occurs in the same way in trajectory models).

5. Page 3, line 3: *The T1279L137 ECMWF operational analysis data used here have 16 km effective horizontal resolution, about 180 − 750m vertical resolution at 2 − 32 km altitude, and are provided at 3 h synoptic time intervals*. "Provided at 3 h . . . " is not entirely correct – it is your choice. Analyses are available every 6 h

and forecasts at steps of 1 h. It would be useful if you indicate what composite of AN and FC fields you were using here and not on the next page.

6. Page 3, line 7: *LPDM studies using this new data set*. It is not clear what you mean by "this new data set". Obiously, ECMWF rules will not allow to make the ECMWF data set that you have used here available for general use.

7. Page 3, line 26: *meteorological wind fields*. Just 'wind fields' should good enough. If the model uses other fields as well (e.g, thermodynamic or surface fields), please explain in more detail. I am also wondering wether the model considers convection – it is invoked as a possible explanation later, but in Hoffmann et al. (2016) I did not find a reference to convection being a simulated process (if it isn't, it should also not be invoked). Maybe you want in general to provide a little bit more information about the model, especially considering that the only paper published so far is not open-access.

8. Page 3, line 31: *While atmospheric reanalyses ... typically have a horizontal resolution of $\sim$ 100 km or less, the resolution of operational forecast products has been continuously improving during the last decades*. Reanaysis products' resolution has improved as well! And better write "$\approx$ 100 km" (\approx) or "ca. 100 km" to not confuse with symbol for proportionality (symbols appear also on p. 7 and p. 10).

9. Page 4, line 4: *For usage with MPTRAC, the wind fields have been interpolated horizontally to a longitude-latitude grid*. Have they really been interpolated (from another, e.g. reduced Gaussian, grid), or were they just extracted at the given grid through MARS (by evaluation of the spectral data)?

10. Math vector notation: You are using upright bold letters for vectors. Standard notation would italic bold, accessible (with the amsmath package) for example through \boldsymbol{text}.

11. Page 6, line 4 ff.: $k_1 = \ldots$. It seems that you define certain velocities as $k$. It is very unusual to denote a velocity by $k$ and not with a letter such as $u$ or $V$, upper- or lowercase, and even more difficult as you don't give an explanation in words of these variables.

12. Page 8, line 5, 8: *5 latitude bands, 3 altitude layers*. According to standard type-setting rules, numbers less or equal to twelve in running text should, in general, be written out (same for "2nd/3rd-order" elsewhere).

13. Page 11, line 30: *land surface ratio*. I guess that "land-surface fraction" is meant.

14. Page 11, line 30: *The tropospheric mid-latitudes were expected to cause the largest errors, because the most complex wind systems occur in this region due to a larger land surface ratio and more complex orography.* The distribution of continents and orography is relevant for the difference between the midlatitudes of the two hemispheres, but not for differences between midlatitudes and else-where – this latter effect is due to the structure of the global circulation which in the end is caused by the poleward increase of the Coriolis parameter, allowing for Rossby waves and baroclinic instability to occur there.

15. Page 12, line 1: *The south pole has the smallest errors*. Probably you want to say that the smallest errors were found over Antarctica / the southern polar region.

16. Page 12, line 5: *The relative high errors in the tropics are probably caused by a stronger turbulence in that region. The lower bound of the stratospheric region of our test cases is 16 km, since the tropopause reaches an average altitude of 16 km near the ITCZ, turbulent movements due to deep convection can occur more frequently in the lower stratosphere above the tropics.* Is turbulence due to convection resolved in MPTRAC? If not, it can't be invoked as an explanation here.

17. Page 12, line 9: *During northern hemisphere wintertime land-sea temperature differences as well as the temperature gradient between the North pole and the equator are largest, which allows for more intense and complex dynamic patterns to occur than in summer.* I would not refer to the meridional temperature gradient as the pole-equator temperature gradient – the pole is a single point and neither the pole nor the equator typically represent the locations of the extreme temperatures. Furthermore, the baroclinicity in mid-latitudes rather depends on the subtropical region temperatures than on equatorial ones.

18. Page 12, line 12: *We need to stress that each simulation lasts only 10 days, which is a relatively short time interval to analyze seasonal effects. Fast temporal variations and changes in medium-range weather patterns can blur out the impact of seasons that is observed here.* To better resolve the seasons you don't need longer trajectories, but more frequent starts or more years. I any case, I don't think that the seasonal effects are so interesting, you could discuss this just briefly. It is obvious that stronger variations in the wind fields will lead to larger truncation errors, and the dependence of the variability of wind fields on the seasons is well known.

19. Page 12, line 27: *Vertical transport deviations are about 800 – 1000 times smaller than the horizontal transport deviations.* As the atmosphere in general is anisotropic ($L \approx 10,000$ km, $H \approx 10$ km), this is trivial and not worth mentioning.

20. Page 12, line 35: *The median error gets somewhat larger in the troposphere, where particle paths are more likely being affected by atmospheric turbulence.* Hoffmann et al. (2016) says that MPTRAC uses the same diffusivity throughout troposphere and stratosphere. How is this compatible?

21. Page 13, line 15: *As an example, Fig. 7 shows results of scaling tests using the midpoint scheme with a time step of 120 s for different numbers of particles and*

*OpenMP threads.* It would be useful to explain why you are only testing OpenMP and a single node if MPTRAC is capable to work on distributed-memory systems as well.

22. Page 13, line 18: *the computing time is limited by an offset of ... s, which is due to the overhead of the OpenMP parallelization.* Language-wise, I would prefer to speak about showing a plateau rather than "being limited by an offeset". Do these times refer only to the time spent in the trajectory calculation, or to the model as a whole? In the latter case, there is not only overhead from parallelisation but also from other parts of the model (the – minor – plateau even with a single a single thread seems to indicate some contribution.) One is also wondering here about your parallelisation strategy – is there a barrier after each time step? Is that needed?

23. Page 13, line 23: *It is also found that the code provides additional speedup if the simultaneous multithreading capabilities of the compute nodes are used, in particular for very large numbers of particles (on the order of $10^6$ to $10^7$). For smaller number of particles ($10^4$ or less) the speedup is limited due to the overhead of the OpenMP parallelization and by the limited work load of the problem itself.* This is an interesting part of your results, but I don't agree completely with your description and interpretation. There is always a drop at first when the number of threads exceeds the number of 24 cores, which is quite typical (see also the indications given in your footnote source). The interesting feature is that for a large enough number of particles, it then rises again. Maybe your computing specialists have more detailed insights for this behaviour? Also, I wondering why for the largest number of particles the first maximum is reached with 20 threads. Is this a plotting error, or is this related to memory access? We should also note some irregular behaviour for moderate numbers of particles toward the maximum number of threads.

24. Page 13, line 33: *Among the 2nd-order methods the Petterssen scheme has the lowest computational efficiency, which is due to the fact that we tuned the convergence criteria for this method for accuracy rather than speed.* So, it is not "the Petterssen scheme" but your implementation of this scheme for which the statement holds! That is a bit of a pity, so we don't know how the Petterssen scheme would do with a more reasonable cut-off of the iterations. As this is quite a relevant issue, and some people might only look at the figure without reading the full thext, I suggest to mention that also in the figure caption (or better do some more realistic tests for a revised version).

25. Page 14, line 1: *The best efficiency, i. e., the best accuracy at the lowest computational costs, is mostly obtained with the midpoint and RK3 methods.* This wording is not providing an operational definition of "best efficiency", as best accuracy and lowest computations cost are mutually exclusive and you are not defining how exactly you want to measure the efficiency. A suitable measure would be the computation time to achieve a given AHTD. Do this for a value that is reasonable and then quantify the computation times, as just reading them out from a log-log diagramme is not so easy (note also the unexplained minor tick intervals better – use a full set of them). Thus, you may want to combine this paragraph with the following one. For the rating of Petterson (vs. midpoint), see above. Another question which needs to be answered is with how many threads this result was obtained, and whether there is any difference between schemes with respect to speed-up.

26. Page 14, line 17: *with an effective horizontal resolution of about 16 km.* Mention also the 3 h here!

27. Page 14, line 18: *The truncation errors of the schemes were found to cluster into three groups that are related to the order of the method.* Add "for a given time step".

28. Page 14, line 25: *We attribute this to larger small-scale variations caused by atmospheric turbulence and mixing in the troposphere*. The first part of the explanation is correct, but the second part not. These variations are not caused by turbulence (16 km is not turbulence scale !!) and certainly not by mixing (this would reduce and not amplify variability!).

29. Page 14/15, line 33–6: *[whole para]*. I suggest to rephrase this paragraph in line with the remarks made above for Section 3.4, making sure it clearly conveys the relevant facts and definitions.

30. Page 15, line 7–9: *The study of Seibert (1993) . . . . To achieve truncation errors that are smaller than overall trajectory uncertainty, they found that the time step should fulfill the CFL criterion as a necessary condition for convergence*. The recommendation there for a sufficiently small truncation error was 15% of the time step needed for convergence of the Petterssen scheme. If we assume that the reference accuracy has also improved in the meantime, an even smaller value would result. The CFL criterion is recommended to make sure that no small-scale features are skipped, not for convergence of the iterations in the Petterssen scheme.

31. Page 15, line 19: *However, the large variability of regional and seasonal truncation errors found here suggests that applications may benefit from more advanced numerical techniques. Adaptive quadrature could be an interesting topic for future research*. Note that adaptive time steps have been recommended by Seibert (1993) and were used already in the 1980ies for atmospheric trajectories by Maryon and Heasman (1988) and Walmsley and Mailhot (1983).

32. References: For Hoppe et al. (2014), quote the final paper and not the discussion version.

33. Figure 1: I would suggest to use the same scale for all pressure levels. I am

wondering why odd pressure levels are used (32.6, 180, 488 hPa) instead of standard levels. And I would suggest to reverse the colour coding for vertical velocity – meteorologists would find it more natural letting blue denote subsidence and red upward motion.

34. Figure 2: I don't deem this figure necessary. If you want to keep it, use an appropriate viewing position in the projection for the Northern hemisphere, presently we are looking from a point located somewhere above the South pole, like peeking through the ground, not down from space! Also, use hollow symbols of different shapes so that we can easily recognise coincident positions as such.

35. Figures 3 ff.: It would help the reader if you annotate subfigures or at least columns of subfigures.

36. Figures 5 and 6: This figure should be simplified. You don't need to show the two years separately, and I think you also don't need to show seasons separately. Thus you could have just three subfigures (three levels) and the five regions inside of each one. Then use a log scale for the AHTD, and symbols instead of bars (which will bring out the median also more clearly).

37. Figures 2, 3, 4, 7, 8: Please make sure that line width, colour intensity and marker size are sufficient to read all the content easily.

38. Using an enlarged printout of the lower part of Fig. 7, I tried to figure out the number of cores which works fastest as a function of the number of particles. I arrived at something like this:

| #particles | #threads | remark |
|---|---|---|
| <50 | 1 | |
| 50 – 200 | 4 | |
| 200 – 300 | 8 | very small interval! |
| 300 – 1000 | 16 | |
| 1,000 – 50,000 | 24 | = #cores! |
| > 50,000 | 48 | = max. #threads |

I think that this evaluation would be useful for users. What is really striking is the fact that only (integer) powers of two show up as recommendable number of threads until 16. Then we can add 8 to arrive at the maximum number of cores (the question is open whether on a 32-core machine, 24 would show up or not), and then we can double once with hyperthreading. This is really a lesson for users, and if you have IT colleagues who are able to relate this behaviour to the hardware layout of your nodes, it would be even more useful.

39. Page 15, line 20: *Code and data availability*.

   • ECMWF data (of the kind used here) are not simply "distributed" by the centre. In general they would be available only for member-state NMS (or institutions authorised by them) and special-project holders. I suggest that the limited availability of these data is indicated. (I also thought that data provision could be mentioned in the acknowledgements.)

   • It would be useful to indicate the availability of the preprocessor which transforms ECMWF data to MPTRAC input data.

   • Does the version of the MPTRAC code available on github include the variety of integration schemes used here? If not, please make a statement about their availability.

   • It would be useful to provide the starting points of the trajectories as supplementary material so that the calculations become more reproducible.

**References**

Bowman, K. P., J. C. Lin, A. Stohl, R. Draxler, P. Konopka, A. Andrews, and D. Brunner (2013), Input data requirements for Lagrangian trajectory models. *Bull. Amer. Meteor. Soc.* **94**, 1051–1058, URL: http://doi.org/10.1175/BAMS-D-12-00076.1.

Brioude, J., W. M. Angevine, S. A. McKeen, and E.-Y. Hsie (2012), Numerical uncertainty at mesoscale in a Lagrangian model in complex terrain. *Geosci. Model Dev.* **5** (5), 1127–1136, URL: http://www.geosci-model-dev.net/5/1127/2012/, doi:10.5194/gmd-5-1127-2012.

Hoffmann, L., T. Rößler, S. Griessbach, Y. Heng, and O. Stein (2016), Lagrangian transport simulations of volcanic sulfur dioxide emissions: Impact of meteorological data products. *J. Geophys. Res.* **121** (9), 4651–4673, URL: http://dx.doi.org/10.1002/2015JD023749.

Maryon, R. and C. Heasman (1988), The accuracy of plume trajectories forecast using the U.K. Meteorological Office operational forecasting models and their sensitivity to calculation schemes. *Atmos. Environ.* **22** (2), 259—-272, URL: http://dx.doi.org/10.1016/0004-6981(88)90032-7.

Seibert, P. (1993), Convergence and accuracy of numerical schemes for trajectory calculation. *J. Appl. Meteorol.* **32**, 558–566.

Stohl, A., G. Wotawa, P. Seibert, and H. Kromp-Kolb (1995), Interpolation errors in wind fields as a function of spatial and temporal resolution and their impact on different types of kinematic trajectories. *J. Appl. Meteorol.* **34**, 2149–2165.

Walmsley, J. L. and J. Mailhot (1983), On the numerical accuracy of trajectory models for long-range transport of atmospheric pollutants. *Atmos.-Ocean* **21** (1), 14—-39, URL: http://dx.doi.org/10.1080/07055900.1983.9649153.
* * *

---

## Referee Comment (RC2) · Anonymous Referee #2 · 21 Mar 2017

**Review for**

**Regional and seasonal truncation errors of trajectory calculations using ECMWF high-resolution operational analyses and forecasts**

**by Rössler et al.**
* * *
**Synopsis:**

In their study the authors look at the truncation errors of six explicit integration schemes of the Runge-Kutta family. The performance is studied based on real-case data from the operational ECMWF analysis and forecasts, whereby the sensitivity with respect to the sphere (troposphere, UTLS, stratosphere) is discussed. Further, the seasonal dependence of the errors is compared, and the computational efficiency is discussed. The paper is very well written, the argumentation very clear, and the number and quality of the figures support well the discussion. I think the paper fits well the interests of the GMD readership, and therefore I certainly can recommend its publication. Still, the authors might want to address the following concerns.

**Concern:**

**1)** The abstract (and manuscript) ends with a rather strong conclusion: "we recommend the 3rd-order Runge Kutta method with a time step of 170 s or the midpoint scheme with a time step of 100 s for efficient simulations of up to 10 days time based on ECMWF's high resolution meteorological data." This is, as the authors note, far below the time step that typically is applied in trajectory calculations based on ECMWF fields. I think the authors can clearly demonstrate that such a small timestep is indeed necessary to get a high degree of accuracy of a *single* trajectory -- where the convergence of the trajectories is assessed based on the AHTD and AVTD distance metric. However, I wonder whether we should trust any single trajectory anyway. Let me make my point more clear: Suppose that we have a calculated a *single* trajectory which reaches after 10 days a AHTD(single) of 100 km. Hence, the trajectory calculation is not perfect. But now also assume that we very slightly change the starting position of the trajectory and repeat the trajectory calculation. We can now compare the distance between the initial and shifted trajectory, and the resulting metric is AHTD(single-shifted) = 200 km. Of course, we could repeat this kind of experiment with several shifted starting positions. The point is that the AHTD(single) can now be seen in a better light, because it is smaller than the inherent spread AHTD(single-shift) due to a minor shift of the starting position. I would argue that the uncertainty of the single trajectory is negligible compared to the flow-inherent dispersion of the trajectories.

In short, I think that there is not too much meaning in considering single trajectories at all. We always have to look at an ensemble of trajectories started from nearby positions. The

coherence of this trajectory ensemble then defines the time horizon until the trajectory is meaningful. Of course, there is also some subjectivity in this argument: The slight shift in starting positions has to be specified. Still, I think the authors should comment on this 'coherent trajectory bundle vs. single trajectory' concept.

**2) In** Figure 5 the winter 2015 stands out. The authors find a reasonable explanation for it: a sudden stratospheric warming and near splitting of the polar vortex. I think this explanation makes perfect sense, and actually points to a potentially interesting extension of the study. In fact, we can expect a varying degree of inter-annual variability not only in the stratosphere, but also in the troposphere and in the UT/LS. There are years with more or less cyclones passing along the storm tracks; there are years where the jet stream in the UT/LS meanders more than in other years (with a more zonal jet). This variability is reflected in climate indices (e.g., the NAO), but it could also be assessed by explicitly 'counting' the cyclones, anticyclones, or by considering a measure of jet zonality. In short, it would be rather interesting to see the trajectory accuracy in context of this inherent tropospheric, UT/LS, and stratospheric flow variability. I don't expect the authors to do that all in the current study! But, possibly they can think about it, and thus link their findings more to meteorology than 'abstract' statistical measures. If appropriate, I would appreciate if the authors comment on this perspective in their study.

**Minor comments:**

**-P2,L25: "**However, it needs to be stressed that appropriate ..." -> "However, the appropriate ..."

**- P3,L1,3,4:** Three sentences starting with 'We' - please rephrase!

**- P4,L12:** "Wind dynamics in the extratropical summer hemisphere are generally slow" -> Unclear what is meant by this statement? Do you want to say that winds in summer are slower? Or that they are not changing as much?

**- P6,L27:** "we calculated the horizontal distances as Cartesian distances of the air parcel positions projected to the Earth surface" -> It is not completely clear how the distance is calculated. What are Cartesian distances on a sphere?

**- P8,L15**: "it needs to be pointed out that this is undersampling" -> "this is still undersampling"

**- Figure 1:** In the upper-right panel the vertical wind velocity is shown. A rather large-scale wave pattern is discernible over northern Europe. I wonder whether this pattern is physical, or some kind of numerical artefact? The amplitude of the waves is rather small, as expected in the stratosphere.

**- Figure 5,6:** I wonder whether it would be better to reduce the number of panels, e.g., by only showing the results for the northern hemisphere? Of course, it would (for instance) also be interesting to compare the northern UT/LS with the southern UT/LS. But, at the moment

the UT/LS is defined by means of fixed heights (8-16 km) and it is not clear whether the tropospheric fraction for the southern hemisphere is the same as for the northern hemisphere. If not, and this will certainly be the case to some degree, the two hemispheres are not really comparable.

**- P11,L1-10:** The values listed in the text are better presented as a table.

**- P11,L29-30:** "The tropospheric mid-latitudes were expected to cause the largest errors, because the most complex wind systems occur in this region due to a larger land surface ratio and more complex orography" -> What do you mean with 'complex wind systems'? What is the 'land surface ratio' - most likely you mean 'land-sea ratio'? Further, it is rather unspecific to attribute the flow variability to the orography and/or the land-to-sea fraction.

**- P11,L33:** Again, what is a 'complex wind pattern' and why is the turbulence higher in this region? I guess  that the authors point to the higher jet variability, i.e., its north-south meandering structure. I would suggest to add some references to climatologies that quantify this variability.

**- P12, L17:** "As a rough indication for inter-annual variability" -> This is indeed a very rough measure for *inter-annual variability*! If I hear 'interannual variability', I would expect a study covering at least 10 years. Hence, I would simply say that the two years 2014 and 2015 are compared, and that they differ substantially -- indicating that the inherent flow properties have a considerable impact on the outcome.

---

## Author Comment (AC1) · 5 May 2017

**Reply to review comments**

We thank the reviewers for the time and efforts spent on the manuscript. We considered all comments and hope that the revised draft properly addresses the remaining issues. Please find our point-by-point replies below (colored in blue).

**Reviewer #1**

1 General

This paper investigates the performance of a number of numerical schemes to integrate the trajectory equation. This is done using the LPDM MPTRAC. As the code is prepared for parallel computing, the performance is investigated also as a function of the number of threads (for one of the schemes only). For the tests, 10-day simulations were carried out using ECMWF data with 16 km grid spacing, and results are presented for different regions of the globe, layers of the atmosphere, and seasons. This study is a useful addition to the previous investigations, as it also tests higher-order methods rarely used in atmospheric transport modelling, and as parallel performance is included. I recommend publishing it after consideration of the following remarks. I think that the authors have some choices with respect to doing additional calculations and/or evaluations, and I hope that they would be able to consider my respective suggestions, as the value of this work could be significantly increased in that way.

2 Major remarks

1. In my opinion, there is one aspect in the setting of the numerical experiments which is not ideal. The regular LPDM code has been used, that is, including a stochastic wind component to represent turbulence. Existing similar studies have been carried out with simple trajectory models. It is not very clear what the consequence of adding stochastic wind components is for the deviations between the schemes tested. The authors propose turbulence as explanation for several of the observed variations in accuracy, but this remains hypothetic. I would strongly recommend to repeat at least a subset of the simulations with all kinds of stochastic influences (turbulence, mesoscale fluctuations, convection if it exists in the model) switched off, present and discuss these results as well.

The simulations are based on the advection module of MPTRAC solely, the modules for turbulence and mesoscale fluctuations were turned off. We added this information to the model description.

2. Another open question is whether RK4 with 60 s time step is a suitable reference method. If one extrapolates the RK3 or RK4 curves in Fig. 8 (bottom), one would arrive at an AHTD value of about 100 km at 60 s (probably against a hypothetical perfect simulation). The time step has to be reduced until a further reduction does not reduce

AHTD significantly in order to establish a reference simulation. (I see that Hoffmann et al. (2016) claim that convergence already was reached at 120 s, but this is in obvious contradiction with the results reported here.) This might change the apparent relative benefits of higher-order methods.

Fig. 8 (bottom) illustrates the convergence for the troposphere, where truncation errors are higher than elsewhere. In fact, the northern mid latitudes slow down the tropospheric convergence. We agree that the AHTD for this particular region suggests that a shorter time step of 30 s might be a better choice. However, the other regions and especially the combined set of all parcels show convergence already for a time step of 60 seconds. The convergence analysis of Hoffmann et al. (2016) is not applicable to this study for two reasons: First, the horizontal resolution was increased from 0.25° to 0.125° in this study, which reduces the convergence rate. Second, the simulations of volcanic emission dispersion by Hoffmann et al. (2016) covered only the UT/LS region, and the results cannot be generalized to the troposphere.

3. As the authors rightly point out, higher-order methods are unlikely to bring much gain if we use linear interpolation. This points to another option for a potentially optimal trajectory calculation, at least as a reference method: Linear interpolation should allow to solve the trajectory equation analytically within a grid cell and between two times of wind field availability (cf. Seibert, 1993). Admittedly, the need to bound each calculation step at grid-cell borders has a potential to make this method a bit cumbersome and computationally probably less efficient.

A reference simulation using linear interpolation would indeed be suited as reference. As our computations do not use such a method, this would also potentially allow for a more solid comparison. However, implementation is complicated due to the implied transformation of spherical and Cartesian coordinates and computation costs are relatively high, so we decided to keep the RK4 reference for the current study.

4. Another methodological issue is the questions on which transport times the final evaluation of schemes should be based. Even though not explicitly mentioned, Fig. 8 seems to be made with results after 10 days. I dont think this is the most appropriate choice. As discussed in Sect. 3.2, there is a strongly non-linear growth of the deviations with time. This growth has nothing to do with numerical errors, it is solely a function of atmospheric flow patterns (diffluent flows or bifurcations). Thus, a longer calculation mainly amplifies initial deviations which are due to the different truncation errors. The longer calculations only mean more calculational efforts, and the true truncation errors are obscured by the increasingly important atmospheric flow influences, probably exaggerating the difference between atmospheric regions or seasons (note also that for example polar-region trajectories mostly leave the polar domain within the 10 days). Please also look at results with much shorter transport times and consider replacing the 10-day results by them.

Our intention was to give estimates for the total uncertainties of trajectory calculations

[Figure]

Figure 1: The figures show the efficiency of the used methods by relating the average error in specific altitudes after 24 h to the required computational time. The dots indicate the time steps from 3600 s on the left to 120 s on the right.

in different atmospheric conditions. We defined a limit for the spread of the parcels and wanted to find the cheapest method to adhere to the limit. However, to allow for distinction between initial truncation errors and those potentially perturbed by atmospheric influence, we added a Figure for the trade-off between computational accuracy and CPU-time after 1 day and discussed the results in Sect. 3.4 (see Figure 1 in this reply).

5. Finally, the results are certainly sensitive to the resolution of the wind field data. Results obtained for the specific case of 16 km / 3 h therefore cannot be generalized. Keeping in mind the conclusions of Stohl et al. (1995), Brioude et al. (2012), and Bowman et al. (2013), 3 h intervals for the wind fields are coarser than what would be desired at this horizontal resolution. As 1 h is provided by ECMWF, I am wondering why it was not used. This also diminishes the value of the results presented here, as most people would want to use the 1-h data if they go to the highest horizontal resolution. There would be a number of ways to produce more general results, such as trying out different resolutions or to parameterize the recommended time step by flow field properties such as (local) spatial and/or temporal derivatives at different orders.

Indeed, hourly operational forecast data can be downloaded from ECMWF since November 2011. However, the description of the operational products at `http://www.ecmwf.int/en/forecasts/datasets/set-i` implies 3-hourly forecast time steps for the first 144 hours of HRES operational forecasts. Similarly, the most recent (updated 2015) user guide to ECMWF forecast products available at `http://www.ecmwf.int/sites/default/files/User_Guide_V1.2_20151123.pdf` specifies the temporal retrieval of ECMWF forecasts as follows: 'All forecast parameters, both surface and upper air, based on 00 and 12 UTC

HRES and ENS, are available at 3-hourly intervals up to +144 hours and at 6-hourly intervals from +150 to +240 hours.' For the scope of this paper we decided to restrict ourselves to data with original and officially approved resolution only and therefore downloaded the operational forecasts with a forecast time step of three hours.

Specific and Minor Remarks

1. The title could be rephrased for example as 'Truncation errors of trajectory calculations using ECMWF high-resolution data diagnosed with the MPTRAC Lagrangian particle dispersion model'

Following the suggestion we rephrased the title of the manuscript to: 'Domain specific truncation errors of trajectory calculations using ECMWF high-resolution data diagnosed with the MPTRAC Lagrangian particle dispersion model'.

2. Page 1, line 1: Abstract. The abstract could be shortened by removing nonessential background and more concise wording.

The abstract has been shorted by removing some unnecessary or redundant background information.

3. Page 1, line 4: kinematic equation of motion (comes also in other places). I dont feel comfortable with this wording. 'Equations of motion' for me would refer to the Euler or Navier-Stokes equations. Why not call this the trajectoy equation?

We think that the term 'kinematic equation of motion' is correctly used for Eq. (1). We do not intend to change the wording.

4. Page 2, line 6: Lagrangian particle dispersion models have proven. Under this chapeau, next to real LPDMs, LAGRANTO is listed which is a simple trajectory model and not an LPDM. I think it does no harm to enumerate it here, but not under a category that doesnt fit (and there is no reason to focus specifically on LPDMs here, as the truncation error problem occurs in the same way in trajectory models).

Our focus is on LPDMs, which is now also visible in the updated title of the study. Therefore we decided to skip the reference to LAGRANTO in the introduction but changed slightly in our conclusions: *Page 14, line 32: All integration methods discussed here are in principle suited and have been used for Lagrangian Particle dispersion and trajectory model simulations.*

5. Page 3, line 3: The T1279L137 ECMWF operational analysis data used here have 16 km effective horizontal resolution, about 180 - 750m vertical resolution at 2 - 32 km altitude, and are provided at 3 h synoptic time intervals. 'Provided at 3 h ... ' is not entirely correct - it is your choice. Analyses are available every 6 h and forecasts at steps of 1 h. It would be useful if you indicate what composite of AN and FC fields you were using here and not on the next page.

ECMWF analyses are produced every 6 hours, but forecasts are only calculated from

the analysis base times of 00 and 12 UTC. Thus we decided to use analyses at 00 and 12 UTC and the corresponding forecasts in between, as described on page 4, lines 5-6. To our opinion such detailed information does not belong to the introduction. Concerning the 1 h forecast steps, please refer to our reply to major remark #5.

6. Page 3, line 7: LPDM studies using this new data set. It is not clear what you mean by 'this new data set'. Obviously, ECMWF rules will not allow to make the ECMWF data set that you have used here available for general use.

*We rephrased the sentence:* Page 3, lines 6-7: Using most recent meteorological data, the results will be of interest for many current and future LPDM studies using ECMWF operational data or data sets with comparable resolution.

7. Page 3, line 26: meteorological wind fields. Just wind fields should good enough. If the model uses other fields as well (e.g, thermodynamic or surface fields), please explain in more detail. I am also wondering whether the model considers convection – it is invoked as a possible explanation later, but in Hoffmann et al. (2016) I did not find a reference to convection being a simulated process (if it isnt, it should also not be invoked). Maybe you want in general to provide a little bit more information about the model, especially considering that the only paper published so far is not open-access.

We rephrased this as suggested. Our model does not consider convection. In the text we refer to convection patterns visible in the meteorological input data. Note that more information on the MPTRAC model can also be found in Heng et al. (2016), which is referenced in our manuscript.

8. Page 3, line 31: While atmospheric reanalyses... typically have a horizontal resolution of $\sim 100\,\mathrm{km}$ or less, the resolution of operational forecast products has been continuously improving during the last decades. Reanalysis products resolution has improved as well! And better write '$\approx 100$ km' (\approx) or 'ca. 100 km' to not confuse with symbol for proportionality (symbols appear also on p. 7 and p. 10).

While it is true that also the resolution of global reanalyses has been improved over time, this has not been done as often as for the operational products. E.g., from ERA-15 (1996) to ERA-INTERIM (2011) the resolution of the ECMWF reanalyses has improved from 1.125° to 0.7° and from 31 to 60 vertical levels, while for the atmospheric operational analyses the resolution has improved from 0.56° to 0.14° and from 31 to 91 levels over the same time frame. Symbols for approximation have been changed throughout the text.

9. Page 4, line 4: For usage with MPTRAC, the wind fields have been interpolated horizontally to a longitude-latitude grid. Have they really been interpolated (from another, e.g. reduced Gaussian, grid), or were they just extracted at the given grid through MARS (by evaluation of the spectral data)?

Wind data on model levels have been directly extracted from MARS by indicating the desired horizontal resolution. The interpolation on pressure levels has been performed by using the model to pressure level interpolation operator ml2pl from the Climate Data

Operators (CDO).

10. Math vector notation: You are using upright bold letters for vectors. Standard notation would italic bold, accessible (with the amsmath package) for example through \boldsymbol{text}.

The notation has been changed accordingly.

11. Page 6, line 4 ff.: k1= ... It seems that you define certain velocities as k. It is very unusual to denote a velocity by k and not with a letter such as u or V, upper-or lowercase, and even more difficult as you dont give an explanation in words of these variables.

The vectors $k_i$ are just auxiliary vectors at different nodes of the integration schemes, for which 'k' may be an acceptable choice of notation. The definitions of these vectors in Eqs. (7) to (15) make clear that wind vectors are meant. Calling the vectors u or v may cause confusion with the wind function that is already called $\boldsymbol{v}$. We kept this as is.

12. Page 8, line 5, 8: 5 latitude bands, 3 altitude layers. According to standard typesetting rules, numbers less or equal to twelve in running text should, in general, be written out (same for '2nd/3rd-order' elsewhere).

This has been fixed throughout the manuscript.

13. Page 11, line 30: land surface ratio. I guess that 'land-surface fraction' is meant.

This is correct. We changed the text accordingly.

14. Page 11, line 30: The tropospheric mid-latitudes were expected to cause the largest errors, because the most complex wind systems occur in this region due to a larger land surface ratio and more complex orography. The distribution of continents and orography is relevant for the difference between the mid-latitudes of the two hemispheres, but not for differences between mid-latitudes and elsewhere - this latter effect is due to the structure of the global circulation which in the end is caused by the poleward increase of the Coriolis parameter, allowing for Rossby waves and baroclinic instability to occur there.

We would like to pick up the remarks of both reviewers to clarify our view on the errors occurring in the mid latitudes. In the original manuscript a hint to the meandering jet streams and the baroclinic structure of the atmosphere was missing, which is a important source of transport errors in our simulations for the mid latitudes. Text on page 11 has been changed as follows: *Page 11, lines 27-30: The troposphere has its largest errors at northern mid-latitudes with errors between 245 km and 470 km. Tropospheric mid-latitudes were expected to cause relatively large errors because of the nature of global circulation: Rossby waves and baroclinic instability occurring predominantly in this region come along with highly variable wind patterns. In addition, the evolution of northern mid-latitudes meteorological systems is more difficult to simulate than for the southern mid-latitudes due to the larger land-sea ratio and more complex orography of the northern hemisphere. The errors in the polar regions... Page 11, lines 32-34: The UT/LS region has its largest AHTDs in the northern mid-latitudes with 95 km to 177 km. These errors are caused by*

*the north-south meandering of the jet (Woollings et al., 2014) and higher turbulence in the underlying region. The second largest...*

15. Page 12, line 1: The south pole has the smallest errors. Probably you want to say that the smallest errors were found over Antarctica / the southern polar region.

We replaced 'South Pole' by 'Antarctica'.

16. Page 12, line 5: The relative high errors in the tropics are probably caused by a stronger turbulence in that region. The lower bound of the stratospheric region of our test cases is 16 km, since the tropopause reaches an average altitude of 16 km near the ITCZ, turbulent movements due to deep convection can occur more frequently in the lower stratosphere above the tropics. Is turbulence due to convection resolved in MPTRAC? If not, it cant be invoked as an explanation here.

The term 'turbulence' was misleading in this context, we intended to refer to the grid-scale fluctuations that are given in the meteorological input data.

17. Page 12, line 9: During northern hemisphere wintertime land-sea temperature differences as well as the temperature gradient between the North pole and the equator are largest, which allows for more intense and complex dynamic patterns to occur than in summer. I would not refer to the meridional temperature gradient as the pole-equator temperature gradient – the pole is a single point and neither the pole nor the equator typically represent the locations of the extreme temperatures. Furthermore, the baroclinicity in mid-latitudes rather depends on the subtropical region temperatures than on equatorial ones.

On page 12, lines 8-10, we replaced 'North Pole' by 'Arctic' and 'Equator' by 'subtropical regions'.

18. Page 12, line 12: We need to stress that each simulation lasts only 10 days, which is a relatively short time interval to analyze seasonal effects. Fast temporal variations and changes in medium-range weather patterns can blur out the impact of seasons that is observed here. To better resolve the seasons you dont need longer trajectories, but more frequent starts or more years. I any case, I dont think that the seasonal effects are so interesting, you could discuss this just briefly. It is obvious that stronger variations in the wind fields will lead to larger truncation errors, and the dependence of the variability of wind fields on the seasons is well known.

We skipped the term 'seasonal truncation errors' from the title of the revised manuscript. Consequently, we changed the title of Sect. 3.3 to 'Regional and temporal truncation errors' in order to include both seasonal and intra-annual effects. The section on seasonal dependencies itself is already very short.

19. Page 12, line 27: Vertical transport deviations are about 800 - 1000 times smaller than the horizontal transport deviations. As the atmosphere in general is anisotropic (L $\approx$ 10,000 km, H $\approx$ 10 km), this is trivial and not worth mentioning.

We omitted this as suggested.

20.  Page 12, line 35: The median error gets somewhat larger in the troposphere, where particle paths are more likely being affected by atmospheric turbulence. Hoffmann et al. (2016) says that MPTRAC uses the same diffusivity throughout troposphere and stratosphere. How is this compatible?

See reply to major remark #1.

21. Page 13, line 15: As an example, Fig. 7 shows results of scaling tests using the midpoint scheme with a time step of 120 s for different numbers of particles and OpenMP threads. It would be useful to explain why you are only testing OpenMP and a single node if MPTRAC is capable to work on distributed-memory systems as well.

We added the following sentence in Sect. 3.4: *The MPI parallelization is only used for ensemble simulations, which are conducted independently on multiple nodes. Therefore, the scalability of the MPI parallelization is mostly limited by I/O issues, which are out of scope of this study.*

22. Page 13, line 18: the computing time is limited by an offset of ... s, which is due to the overhead of the OpenMP parallelization. Language-wise, I would prefer to speak about showing a plateau rather than 'being limited by an offset'. Do these times refer only to the time spent in the trajectory calculation, or to the model as a whole? In the latter case, there is not only overhead from parallelization but also from other parts of the model (the minor plateau even with a single a single thread seems to indicate some contribution.) One is also wondering here about your parallelization strategy – is there a barrier after each time step? Is that needed?

The time measurements refer only to the part of the code spent in the advection module of MPTRAC. Due to the operator splitting approach used by our model, an OpenMP barrier occurs after the call of each operator (or 'module' of MPTRAC) and after each time step. Future work may focus on 'pipelining' of the operators, but this would require a major revision of the structure of our model. We will replace the word 'offset' by 'constant contribution that can be attributed to the OpenMP parallelization overhead'.

23. Page 13, line 23: It is also found that the code provides additional speedup if the simultaneous multithreading capabilities of the compute nodes are used, in particular for very large numbers of particles (on the order of $10^6$ to $10^7$). For smaller number of particles ($10^4$ or less) the speedup is limited due to the overhead of the OpenMP parallelization and by the limited work load of the problem itself. This is an interesting part of your results, but I dont agree completely with your description and interpretation. There is always a drop at first when the number of threads exceeds the number of 24 cores, which is quite typical (see also the indications given in your footnote source). The interesting feature is that for a large enough number of particles, it then rises again. Maybe your computing specialists have more detailed insights for this behaviour? Also, I was wondering why for the largest number of particles the first maximum is reached with 20 threads. Is this a

plotting error, or is this related to memory access? We should also note some irregular behaviour for moderate numbers of particles toward the maximum number of threads.

We consulted the IT experts at our center to get more information. According to their analysis, limited scalability (or 'drops' in speed-up) can be assigned to load imbalances. Our model implicitly uses a 'static' schedule for the OpenMP loop parallelization. For instance, for $10^6$ particles on 28 threads there will be 4 cores that have to process two packages of $36\,\mathrm{k}$ particles using hyper threading (HT) while the other 20 cores only process one package without HT. This implies a significant load imbalance compared to a more balanced scaling using 24 threads, which corresponds to the number of physical cores. Nevertheless, speedup results at 48 threads compared to 24 threads show that running with HT is 45% more efficient than without.

24. Page 13, line 33: Among the 2nd-order methods the Petterssen scheme has the lowest computational efficiency, which is due to the fact that we tuned the convergence criteria for this method for accuracy rather than speed. So, it is not 'the Petterssen scheme' but your implementation of this scheme for which the statement holds! That is a bit of a pity, so we dont know how the Petterssen scheme would do with a more reasonable cut-off of the iterations. As this is quite a relevant issue, and some people might only look at the figure without reading the full text, I suggest to mention that also in the figure caption (or better do some more realistic tests for a revised version).

The Petterssen scheme with many iterations did not give significantly more accurate results than the second order methods, which include Heun's method, which is equal to the Petterssen scheme with one iteration. Therefore no further analysis of intermediate configurations was made. However, we share your concern and added the following note to the caption of Fig. 8: *Note that our implementation of the Petterssen scheme was optimized for numerical accuracy rather than speed.*

25. Page 14, line 1: The best efficiency, i. e., the best accuracy at the lowest computational costs, is mostly obtained with the midpoint and RK3 methods. This wording is not providing an operational definition of 'best efficiency', as best accuracy and lowest computations cost are mutually exclusive and you are not defining how exactly you want to measure the efficiency. A suitable measure would be the computation time to achieve a given AHTD. Do this for a value that is reasonable and then quantify the computation times, as just reading them out from a log-log diagram is not so easy (note also the unexplained minor tick intervals better use a full set of them). Thus, you may want to combine this paragraph with the following one. For the rating of Petterssen (vs. midpoint), see above. Another question which needs to be answered is with how many threads this result was obtained, and whether there is any difference between schemes with respect to speed-up.

The most efficient method was detected as suggested by the reviewer, and this has been made more clear in a revision of this paragraph. The computation used 48 cores and the methods profit differently from the parallelization. Figure 2 in this reply shows the relative

[Figure]

Figure 2: Relative computation time of the methods. The estimated times are based on the assumption that the time scales linearly with the number of calls to the wind interpolation function.

computation time of the methods in comparison to the Euler method. Theoretically there should be a linear dependency between computation time and the number of calls to the wind interpolation function (which is the most expensive part of the advection module). However, the higher order methods, which call to the wind interpolation more often, are faster than this estimate for the computational time. Note that the maximum number of iterations for the Petterssen scheme was six, which explains the plateau. We contribute the better speedup for more computationally expensive methods to cache usage, since the wind interpolation probably considers some grid points more than once, such that elements can be read from the cache instead from main memory. However, this makes the RK4 and Petterssen scheme even less attractive, since the computation time would be even larger without higher parallelization speedup.

26. Page 14, line 17: with an effective horizontal resolution of about 16 km. Mention also the 3 h here!

The information about the temporal resolution has been added accordingly.

27. Page 14, line 18: The truncation errors of the schemes were found to cluster into three groups that are related to the order of the method. Add 'for a given time step'.

This has been added accordingly.

28. Page 14, line 25: We attribute this to larger small-scale variations caused by atmospheric turbulence and mixing in the troposphere. The first part of the explanation

is correct, but the second part not. These variations are not caused by turbulence (16 km is not turbulence scale !!) and certainly not by mixing (this would reduce and not amplify variability!).

*We omitted the wrong part of the explanation.*

29. Page 14/15, line 336: [whole para]. I suggest to rephrase this paragraph in line with the remarks made above for Sect. 3.4, making sure it clearly conveys the relevant facts and definitions.

*The paragraph has been rewritten taking into account your remarks #21 to #25.*

30. Page 15, line 7-9: The study of Seibert (1993)... . To achieve truncation errors that are smaller than overall trajectory uncertainty, they found that the time step should fulfill the CFL criterion as a necessary condition for convergence. The recommendation there for a sufficiently small truncation error was 15% of the time step needed for convergence of the Petterssen scheme. If we assume that the reference accuracy has also improved in the meantime, an even smaller value would result. The CFL criterion is recommended to make sure that no small- scale features are skipped, not for convergence of the iterations in the Petterssen scheme.

*We adjusted the paragraph according to your comment and added the following: Page 15, line 9: Their recommendation for a sufficiently small truncation error was 15% of the time step needed for convergence of the Petterssen scheme. Assuming that the reference accuracy has improved in the meantime, an even smaller value would result. The CFL criterion is used to make sure that no small-scale features are skipped.*

31. Page 15, line 19: However, the large variability of regional and seasonal truncation errors found here suggests that applications may benefit from more advanced numerical techniques. Adaptive quadrature could be an interesting topic for future research. Note that adaptive time steps have been recommended by Seibert (1993) and were used already in the 1980ies for atmospheric trajectories by Maryon and Heasman (1988) and Walmsley and Mailhot (1983).

*We made a reference to the mentioned studies: Page 15, lines 18-19: However, the large variability of regional and seasonal truncation errors found here suggests that applications may benefit from more advanced numerical techniques. Adaptive time stepping as recommended by Seibert (1993) was used already in the 1980s for atmospheric trajectories by Maryon and Heasman (1988) and Walmsley and Mailhot (1983). Such an adaptive quadrature could be taken up for future research.*

32. References: For Hoppe et al. (2014), quote the final paper and not the discussion version.

*The reference has been updated in the final manuscript.*

33. Figure 1: I would suggest to use the same scale for all pressure levels. I am wondering why odd pressure levels are used (32.6, 180, 488 hPa) instead of standard levels.

[Figure]

Figure 3: Examples of trajectory calculations using different numerical integration schemes. Circles mark the starting positions of the trajectories. Trajectories were launched at an altitude of 10.8 km (left) and 9.7 km (right). The starting time is 1 January 2014, 00:00 UTC in both examples. Triangles mark trajectory positions at 0 UTC on each day.

And I would suggest to reverse the colour coding for vertical velocity – meteorologists would find it more natural letting blue denote subsidence and red upward motion.

For better visibility of the circulation patterns we decided to use different scales for the three pressure levels. In case of vertical velocities the maximum values differ by more than a magnitude between the levels. The chosen pressure levels are used in the model and correspond closely to the altitudes given in the figure caption. The colour coding for vertical velocity has been reversed following comments by reviewer #2.

34. Figure 2: I dont deem this figure necessary. If you want to keep it, use an appropriate viewing position in the projection for the Northern hemisphere, presently we are looking from a point located somewhere above the South pole, like peeking through the ground, not down from space! Also, use hollow symbols of different shapes so that we can easily recognize coincident positions as such.

We corrected the projection error and changed the symbols (see Figure 3).

35. Figures 3 ff.: It would help the reader if you annotate subfigures or at least columns of subfigures.

This will be done during copy-editing.

36. Figures 5 and 6: This figure should be simplified. You dont need to show the two years separately, and I think you also dont need to show seasons separately. Thus you could have just three subfigures (three levels) and the five regions inside of each one. Then

[Figure]

Figure 4: Average and median horizontal transport deviations after 10 days in different regions for the RK3 method. The orange horizontal line represents the average of the domain. The gray horizontal line indicates the error limit.

use a log scale for the AHTD, and symbols instead of bars (which will bring out the median also more clearly).

We would like to show the simulation results separately because regional and temporal impacts on the error were a part of the motivation for this study. We added a horizontal line for the average error to all subfigures (see Figures 4 and 5). We did not use a log scale, because it would hide the seasonal and regional differences.

37. Figures 2, 3, 4, 7, 8: Please make sure that line width, colour intensity and marker size are sufficient to read all the content easily.

We tried to improve the figures accordingly.

38. Using an enlarged printout of the lower part of Fig. 7, I tried to figure out the number of cores which works fastest as a function of the number of particles. I arrived at something like this:

**particles #threads remark**

<50 1

50 - 200 4

[Figure]

Figure 5: Average and median vertical transport deviations after 10 days in different regions for the RK3 method. The orange horizontal line represents the average of the domain.

200 - 300 8 very small interval!

300 - 1000 16

1,000 - 50,000 24 = #cores!

> 50,000 48 = max. #threads

I think that this evaluation would be useful for users. What is really striking is the fact that only (integer) powers of two show up as recommendable number of threads until 16. Then we can add 8 to arrive at the maximum number of cores (the question is open whether on a 32-core machine, 24 would show up or not), and then we can double once with hyper-threading. This is really a lesson for users, and if you have IT colleagues who are able to relate this behaviour to the hardware layout of your nodes, it would be even more useful.

This is a helpful evaluation and we added a statement in the paper summarizing the findings regarding the number of threads providing the minimum computation time with respect to the number of particles. Unfortunately, our IT experts were not able to provide a simple explanation of how the number of threads is linked to the hardware layout. The findings may depend specifically on the computing architecture and should not be generalized too much.

39. Page 15, line 20: Code and data availability. - ECMWF data (of the kind used here) are not simply 'distributed' by the centre. In general they would be available only for member-state NMS (or institutions authorized by them) and special-project holders. I suggest that the limited availability of these data is indicated. (I also thought that data provision could be mentioned in the acknowledgements.) - It would be useful to indicate the availability of the preprocessor which transforms ECMWF data to MPTRAC input data. - Does the version of the MPTRAC code available on github include the variety of integration schemes used here? If not, please make a statement about their availability. - It would be useful to provide the starting points of the trajectories as supplementary material so that the calculations become more reproducible.

There are several options to obtain ECMWF operational data, all of them are described in http://www.ecmwf.int/en/forecasts/accessing-forecasts. We crated a separate repository containing the MPTRAC code for the various integration schemes as well as the starting points of the trajectories. Section 5 has been changed as follows: *Page 15, lines 18-19: Operational analyses and forecasts can be obtained from the European Centre for Medium-Range Weather Forecasts (ECMWF), see http://www.ecmwf.int/en/forecasts (last access: 3 May 2017) for further details on data availability and restrictions. ECMWF data have been processed for usage with MPTRAC by means of the Climate Data Operators (CDO, https://code.zmaw.de/projects/cdo, last access: 3 May 2017). The version of the MPTRAC model that was used for this study along with the model initializations is available under the terms and conditions of the GNU General Public License, Version 3 from the repository at https://github.com/slcs-jsc/mptrac-advect (last access: 3 May 2017).*

**Reviewer #2**

Synopsis:

In their study the authors look at the truncation errors of six explicit integration schemes of the Runge-Kutta family. The performance is studied based on real-case data from the operational ECMWF analysis and forecasts, whereby the sensitivity with respect to the sphere (troposphere, UTLS, stratosphere) is discussed. Further, the seasonal dependence of the errors is compared, and the computational efficiency is discussed. The paper is very well written, the argumentation very clear, and the number and quality of the figures support well the discussion. I think the paper fits well the interests of the GMD readership, and therefore I certainly can recommend its publication. Still, the authors might want to address the following concerns.

Concern:

1) The abstract (and manuscript) ends with a rather strong conclusion: "we recommend the 3rd-order Runge Kutta method with a time step of 170 s or the midpoint scheme with a time step of 100 s for efficient simulations of up to 10 days time based on ECMWFs high resolution meteorological data." This is, as the authors note, far below the time step that typically is applied in trajectory calculations based on ECMWF fields. I think the authors can clearly demonstrate that such a small timestep is indeed necessary to get a high degree of accuracy of a single trajectory – where the convergence of the trajectories is assessed based on the AHTD and AVTD distance metric. However, I wonder whether we should trust any single trajectory anyway. Let me make my point more clear: Suppose that we have a calculated a single trajectory which reaches after 10 days a AHTD(single) of 100 km. Hence, the trajectory calculation is not perfect. But now also assume that we very slightly change the starting position of the trajectory and repeat the trajectory calculation. We can now compare the distance between the initial and shifted trajectory, and the resulting metric is AHTD(single-shifted) = 200 km. Of course, we could repeat this kind of experiment with several shifted starting positions. The point is that the AHTD(single) can now be seen in a better light, because it is smaller than the inherent spread AHTD(single-shift) due to a minor shift of the starting position. I would argue that the uncertainty of the single trajectory is negligible compared to the flow-inherent dispersion of the trajectories. In short, I think that there is not too much meaning in considering single trajectories at all. We always have to look at an ensemble of trajectories started from nearby positions. The coherence of this trajectory ensemble then defines the time horizon until the trajectory is meaningful. Of course, there is also some subjectivity in this argument: The slight shift in starting positions has to be specified. Still, I think the authors should comment on this 'coherent trajectory bundle vs. single trajectory' concept.

Usual simulations with MPTRAC follow your approach and many parcels are randomly distributed around a starting point. Alternatively, in this study many different starting points are used, such that the impact of the average atmospheric conditions of the domains

on the error can be estimated. The analysis of the error would become very costly, if groups of parcels were created for each starting point. We tried to show that deviations of individual trajectories are not very meaningful by additionally computing the median deviation of the parcels. However, our impression is that the AHTD/AVTD metric is the common method for trajectory evaluation and we wanted to make the results comparable to existing studies. Also, with this setup , we wanted to reach convergence for the trajectories to compare the errors of different methods.

2) In Figure 5 the winter 2015 stands out. The authors find a reasonable explanation for it: a sudden stratospheric warming and near splitting of the polar vortex. I think this explanation makes perfect sense, and actually points to a potentially interesting extension of the study. In fact, we can expect a varying degree of inter-annual variability not only in the stratosphere, but also in the troposphere and in the UT/LS. There are years with more or less cyclones passing along the storm tracks; there are years where the jet stream in the UT/LS meanders more than in other years (with a more zonal jet). This variability is reflected in climate indices (e.g., the NAO), but it could also be assessed by explicitly 'counting' the cyclones, anticyclones, or by considering a measure of jet zonality. In short, it would be rather interesting to see the trajectory accuracy in context of this inherent tropospheric, UT/LS, and stratospheric flow variability. I don't expect the authors to do that all in the current study! But, possibly they can think about it, and thus link their findings more to meteorology than 'abstract' statistical measures. If appropriate, I would appreciate if the authors comment on this perspective in their study.

We would like to thank the reviewer for bringing up this interesting starting point for further research on trajectory accuracy in context of inter-annual variability of tropospheric and stratospheric atmospheric flow. Indeed, we did not expect such large variations between two NH winters in the first place and it would be intriguing to extend such a study to multiple years once input data at sufficient and constant resolution will be available.

Minor comments:

-P2,L25: "However, it needs to be stressed that appropriate ..." → "However, the appropriate ..."

Text has been changed accordingly.

- P3,L1,3,4: Three sentences starting with 'We' - please rephrase!

Text has been rephrased.

- P4,L12: "Wind dynamics in the extratropical summer hemisphere are generally slow" → Unclear what is meant by this statement? Do you want to say that winds in summer are slower? Or that they are not changing as much?

We tried to make this point more clear by saying 'Stratospheric wind speeds in the extratropical summer hemisphere are generally slow compared to the winter hemisphere.'

- P6,L27: "we calculated the horizontal distances as Cartesian distances of the air parcel

[Figure]

Figure 6: Atmospheric InfraRed Sounder (AIRS/Aqua) satellite observations of stratospheric gravity waves (following Hoffmann et al., 2013).

positions projected to the Earth surface" → It is not completely clear how the distance is calculated. What are Cartesian distances on a sphere?

To clarify we rephrased this: 'To calculate the horizontal distances we converted the spherical coordinates of the air parcels to Cartesian coordinates and calculated the Euclidean distance of the Cartesian coordinates.' Note that this approach approximates spherical distances quite well, as long as those distance are smaller than about 3000 km.

- P8,L15: "it needs to be pointed out that this is undersampling" → "this is still undersampling"

We rephrased accordingly.

- Figure 1: In the upper-right panel the vertical wind velocity is shown. A rather large-scale wave pattern is discernible over northern Europe. I wonder whether this pattern is physical, or some kind of numerical artifact? The amplitude of the waves is rather small, as expected in the stratosphere.

Although we can not exclude that some numerical artifacts of the ECMWF IFS model are present in the vertical velocity map, there is evidence that the wave structures are

physical, because they occur in the same places where an infrared nadir sounder observed stratospheric gravity waves. See Figure 6 in this reply.

- Figure 5,6: I wonder whether it would be better to reduce the number of panels, e.g., by only showing the results for the northern hemisphere? Of course, it would (for instance) also be interesting to compare the northern UT/LS with the southern UT/LS. But, at the moment the UT/LS is defined by means of fixed heights (8-16 km) and it is not clear whether the tropospheric fraction for the southern hemisphere is the same as for the northern hemisphere. If not, and this will certainly be the case to some degree, the two hemispheres are not really comparable.

The intention of our Figures 5 and 6 is to give a comprehensive impression of spatial and temporal variability of transport deviations on the global scale. Although the altitude classification namely in the UT/LS region does not exactly reflect the real tropospheric and stratospheric fractions and their hemispheric variations, it still shows substantial differences between the hemispheres.

- P11,L1-10: The values listed in the text are better presented as a table.

We added a new table (Table 2) which comprises the values originally given on page 11, lines 1-8.

- P11,L29-30: "The tropospheric mid-latitudes were expected to cause the largest errors, because the most complex wind systems occur in this region due to a larger land surface ratio and more complex orography" → What do you mean with 'complex wind systems'? What is the 'land surface ratio' - most likely you mean 'land-sea ratio'? Further, it is rather unspecific to attribute the flow variability to the orography and/or the land-to-sea fraction.

We decided to rephrase the whole paragraph and would like to refer to our reply to minor remark #14 of reviewer #1. 'Land surface ratio' has been changed to 'land-sea ratio'.

- P11,L33: Again, what is a 'complex wind pattern' and why is the turbulence higher in this region? I guess that the authors point to the higher jet variability, i.e., its north-south meandering structure. I would suggest to add some references to climatologies that quantify this variability.

Please see our reply to minor remark #14 of reviewer #1.

- P12, L17: "As a rough indication for inter-annual variability" → This is indeed a very rough measure for inter-annual variability! If I hear 'interannual variability', I would expect a study covering at least 10 years. Hence, I would simply say that the two years 2014 and 2015 are compared, and that they differ substantially – indicating that the inherent flow properties have a considerable impact on the outcome.

We share your concern. Although our study used two years to give an initial indication of inter-annual variability, we would rather speak of differences between the two test years

instead of an inter-annual variability. We changed the text accordingly.

**Executive editor comment**

Dear authors,

in my role as Executive editor of GMD, I would like to bring to your attention our Editorial version 1.1:

http://www.geosci-model-dev.net/8/3487/2015/gmd-8-3487-2015.html

This highlights some requirements of papers published in GMD, which is also available on the GMD website in the 'Manuscript Types' section:

http://www.geoscientific-model-development.net/submission/manuscript_types.html

In particular, please note that for your paper, the following requirements have not been met in the Discussions paper:

- The main paper must give the model name and version number (or other unique identifier) in the title.

- If the model development relates to a single model then the model name and the version number must be included in the title of the paper. If the main intention of an article is to make a general (i.e. model independent) statement about the usefulness of a new development, but the usefulness is shown with the help of one specific model, the model name and version number must be stated in the title. The title could have a form such as, "Title outlining amazing generic advance: a case study with Model XXX (version Y)".

So please add the model name and/or its acronym (MPTRAC) and its respective version number in the title of your article in your revised submission to GMD.

Yours, Astrid Kerkweg

We rephrased the title of the manuscript according to suggestions made by reviewer #1. The name of the model, MPTRAC, was included. Unfortunately, a specific version number was not assigned for the code used here. However, to allow others to reproduce our results, we made the code available in a separate repository, as described in the the revised section on 'code and data availability' in our manuscript.

**References**

Heng, Y., Hoffmann, L., Griessbach, S., Rößler, T., and Stein, O.: Inverse transport modeling of volcanic sulfur dioxide emissions using large-scale simulations, Geosci. Model Dev., 9, 1627–1645, 2016.

Hoffmann, L., Xue, X., and Alexander, M. J.: A global view of stratospheric gravity wave hotspots located with Atmospheric Infrared Sounder observations, J. Geophys. Res., 118, 416–434, 2013.

Hoffmann, L., Rößler, T., Griessbach, S., Heng, Y., and Stein, O.: Lagrangian transport simulations of volcanic sulfur dioxide emissions: impact of meteorological data products, J. Geophys. Res., doi: 10.1002/2015JD023749, 2016.

Maryon, R. and Heasman, C.: The accuracy of plume trajectories forecast using the UK Meteorological Office operational forecasting models and their sensitivity to calculation schemes, Atmos. Environment, 22, 259–272, 1988.

Seibert, P.: Convergence and accuracy of numerical methods for trajectory calculations, J. Appl. Met., 32, 558–566, 1993.

Walmsley, J. L. and Mailhot, J.: On the numerical accuracy of trajectory models for long-range transport of atmospheric pollutants, Atmosphere-Ocean, 21, 14–39, 1983.

Woollings, T., Czuchnicki, C., and Franzke, C.: Twentieth century North Atlantic jet variability., Quart. J. Roy. Meteorol. Soc., 140, 783791, doi: 10.1002/qj.2197, 2014.

---

## Referee Report (RR1)

**1 General**

The authors have implemented suggestions and thus improved their paper. However, not all recommendations have been sufficiently addressed, as detailed below. Specifically, certain problems related to meteorological reasoning, for example invoking turbulence as explanation for larger transport errors, have not been resolved.

Quotations from my orginal comments are set in *italics*, those from the authors' answers or revised manuscript in 'single quotes'.

**2 Major remarks**

1. clarified

2. *Another open question is whether RK4 with 60 s time step is a suitable reference method.* The authors accept this critism for the northern midlatitudes. However, they don't offer an improvement or a justification of keeping their reference. I would expect that the authors address this point either by repeating the calculations and evaluations with a shorter reference time step, or by adding an explanation and justification within their paper. Their remark on Hoffman et al. (2016) is not leading to an argument, it just repeats and confirms what I wrote.

3. clarified

4. OK.

5. *Finally, the results are certainly sensitive to the resolution advection module and ECMWF operational analysesof the wind field data. Results obtained for the specific case of 16 km / 3 h therefore cannot be generalised. Keeping in mind the conclusions of Stohl1995, Brioude2012, and Bowman2013, 3 h intervals for the wind fields are coarser than what would be desired at this horizontal resolution. As 1 h is provided by ECMWF, I am wondering why it was not used.*
   I don't feel satisfied by the answer given. It is not appropriate to call the 3 h 'officially approved resolution' (there is no such thing as official approval, all what is in MARS is usable). The question of how results would change with 1 h data is highly pertinent. Alternatively, one could test coarser temporal in combination with coarser spatial resolution.

6. New issue created by introducing the wording 'total error' in recognition of the fact that the transport errors obtained are not pure truncation errors: As shortening the time step eliminates only errors introduced by the truncation error, the 10-day error is not the total error. The total error would be larger than that as it has other contributions as well which are also amplified during the 10-day transport. The authors should make that clear and find another, more appropriate wording. See also major remark #1 of Reviewer #2.

**3 Specific and Minor Remarks**

1. Authors now propose the title 'Domain specific trajectory errors diagnosed with the MP-TRAC advection module and ECMWF operational analyses'. First of all, one would have to hyphenate 'domain-specific'. Then, I think the word *domain* is not the best choice to express that evaluations were done separately for different regions (*domain* usually refers to a calculation domain and not to a climatological region.) Furthermore, MPTRAC and

ECMWF data are not on the same level, one is the model, the other model input. Better say something like 'Trajectory errors as a function of numerical scheme, time step, and region of the atmosphere diagnosed with the MPTRAC model'. Maybe you can add 'for ECMWF wind fields', but I think it is not so important to bring that into the title.

2. Page 1, line 1: *Abstract*. OK

3. *Page 1, line 4: kinematic equation of motion* (comes also in other places). *I don't feel comfortable with this wording. "Equations of motion" for me would refer to the Euler or Navier-Stokes equations. Why not call this the trajectoy equation?*
Authors decline to change their wording without providing arguments. I will accept their wording if they provide a quotation from a well-established meteorological textbook which uses 'eqation of motion' for the kinematic trajectory equation.

4. OK

5. It is not true that no forecasts are produced from 06 and 18 UTC analysis. However, these are not long-term forecasts but just for providing background fields at the next major analysis step.
I accept the argument that these details don't belong to the introduction. Therefore, I would suggest to provide all the information about resolution and kind of ECMWF input fields only in the Section 2, and to remove them completely from Section 1. Otherwise, one is wondering about partial information until one has reached the next section.

6. OK

7. Reformulation is ok. I don't have the impresson that Heng et al. (2016) provides a full model description.

8. OK

9. The explanation is ok but it should also go into the revised manuscript.

10. OK

11. *Page 6, line 4 ff.: $k_1 = \ldots$. It seems that you define certain velocities as $k$.*
I regret that the authors want to keep this notation. In any case, symbols have to be explicitly explained (in words) before or immediately after first usage, whether or not there is an equation which defines them. This is a standard in scientific publications.

12. OK

13. OK

14. *Page 11, line 30: The tropospheric mid-latitudes were expected to cause the largest errors, because the most complex wind systems occur in this region due to a larger land surface ratio and more complex orography.* The authors offer some improvement, but it is still insufficient.
(1) They should pay attention to their style. For example, wordings such as 'The troposphere has its largest error' are not correct (this example is not the only such mistake). The troposphere cannot have any error, only calculations can.
(2) We still find the phrase 'the evolution of northern mid-latitudes meteorological systems is more difficult to simulate than for the southern mid-latitudes due to the larger land-sea ratio and more complex orography of the northern hemisphere.' Apart from the

question whether this is true or not, the difficulty to simulate the evolution of meteorological systems (in other words, the predictability) is not relevant for trajectory errors based on analyses.

(3) 'These errors are caused by ... and higher turbulence in the underlying region'. In their answer to minor remark 16 authors admit that turbulence is not relevant – thus, why do they again come up with turbulence as an explanation of trajectory errors?

15. OK

16. *Page 12, line 5: The relative high errors in the tropics are probably caused by a stronger turbulence in that region. The lower bound of the stratospheric region of our test cases is 16 km, since the tropopause reaches an average altitude of 16 km near the ITCZ, turbulent movements due to deep convection can occur more frequently in the lower stratosphere above the tropics.*

The authors admit that the term 'turbulence' is misleading here, but they have not changed their wording. The explanation by turbulence is wrong and has to be removed. If they want to refer to 'fluctuations in the meteorological input data', I think they have to be more specific what is different in the tropics compared to mid-latitudes. Mid-latitude wind fields also show fluctuations.

17. OK

18. *Page 12, line 12: We need to stress that each simulation lasts only 10 days, which is a relatively short time interval to analyze seasonal effects. Fast temporal variations and changes in medium-range weather patterns can blur out the impact of seasons that is observed here. To better resolve the seasons you don't need longer trajectories, but more frequent starts or more years. I any case, I don't think that the seasonal effects are so interesting, you could discuss this just briefly. It is obvious that stronger variations in the wind fields will lead to larger truncation errors, and the dependence of the variability of wind fields on the seasons is well known.*

Most of this comment is not at all addressed in the reply by the authors. It is really questionable whether seasons are represented in a statistically adequate way with a single day on which trajectories were started. Could you not just add some more? They did 5000 (10-day) trajectory simulations, with a parallelised model on a supercomputer. Doing 20000 or 50000 instead would not be a serious burden in terms of computing work. It is true that the section on seasonal results is not very long, but if we add also the figures, it is also not so short. Generally speaking, I consider the merits of this paper lying in the realm of numerical methods, showing a systematic comparison of a number of numerical integration schemes. The layer of meteorological interpretation according to season, region etc. which has been put around that has much less scientific substance, and in its present formulation even is partly mistaken (see the turbulence issue). It would be better to de-emphasise this part.

19. OK

20. *Page 12, line 35: The median error gets somewhat larger in the troposphere, where particle paths are more likely being affected by atmospheric turbulence. Hoffmann et al 2016 says that MPTRAC uses the same diffusivity throughout troposphere and stratosphere. How is this compatible?*

The authors say that they answered this in their response to major comment 1. However,

there they only explain that turbulence is not active in their simulations. But then, this argument simple collapses; however, the sentence in question has been left unchanged.

21. *It would be useful to explain why you are only testing OpenMP and a single node if MP-TRAC is capable to work on distributed-memory systems as well.* Answer: 'We added the following sentence in Sect. 3.4: The MPI parallelization is only used for ensemble simulations, which are conducted independently on multiple nodes. Therefore, the scalability of the MPI parallelization is mostly limited by I/O issues, which are out of scope of this study.'
I don't understand the argument. If the strategy for distributed-memory machines is trivial parallelisation by multiple runs started concurrently, why does the code offer MPI-based parallelisation?

22. The explanation 'The time measurements refer only to the part of the code spent in the advection module of MPTRAC.' should be included also in the manuscript text. Furthermore, if this is the case, the contribution independent of the number of threads also stems from non-parallel parts of the code, not only from the OpenMP overheads.

23. The explanation given should be included in the manuscript text.

24. As the Pettersson method is widely used, it would really be desirable that the authors test its efficiency with a reasonable iteration cut-off compared for example to the midpoint method.

25. It would be good to put the explanations, e.g. about the role of cache for different numerical schemes, into the manuscript text.

26. OK

27. OK

28. *Page 14, line 25: We attribute this to larger small-scale variations caused by atmospheric turbulence and mixing in the troposphere. The first part of the explanation is correct, but the second part not. These variations are not caused by turbulence (16 km is not turbulence scale !!) and certainly not by mixing (this would reduce and not amplify variability!).*
The authors have removed the reference to mixing, but they keep the reference to turbulence. As I have tried to explain in various parts of the paper where turbulence is invoked, this is not appropriate. Atmospheric turbulence does not create variability at the high-frequency end of the resolved motion scales. It will rather tend to undo existing gradients.

29. *Summary and conclusions*: Please spell out RK where it occurs for the first time in this section (some people may read only this section).
'After 24 h the trajectory errors are quite similar in the troposphere and stratosphere' – Figure 4 shows a difference by about a factor of 10 for AHTD, thus they are not 'quite similar'.
'We attribute this to larger small-scale variations caused by atmospheric turbulence.' – Remove erroneous reference to turbulence.
Statistics not being sufficiently robust: as said before, it would be desirable to increase the sample size.

30. *Page 15, line 7–9: The study of Seibert (1993) . . . . To achieve truncation errors that are smaller than overall trajectory uncertainty, they found that the time step should fulfill the CFL criterion as a necessary condition for convergence. The recommendation there*

*for a sufficiently small truncation error was 15% of the time step needed for convergence
of the Petterssen scheme. If we assume that the reference accuracy has also improved in
the meantime, an even smaller value would result. The CFL criterion is recommended to
make sure that no small-scale features are skipped, not for convergence of the iterations
in the Petterssen scheme.*

The authors have amended their wording, but they have not changed the first sentence
quoted above, which is not an accurate representation of Seibert (1993), as explained in
my previous comment.

31. OK

32. OK

33. 'For better visibility of the circulation patterns we decided to use different scales for the
three pressure levels. In case of vertical velocities the maximum values differ by more
than a magnitude between the levels.'
I think a part of the motivation for showing sample vertical motion patterns at three
different levels is exactly this fact that vertical motions are much smaller in the strato-
sphere, and this is hidden by changing the scale between the figures.
'The chosen pressure levels are used in the model and correspond closely to the altitudes
given in the figure caption.'
If standard pressure levels also are available, it would be preferable to shown them.
'The colour coding for vertical velocity has been reversed following comments by reviewer
**2.'**
Thank you.

34. OK

35. I hope this annotation will really happen – why not doing it now?

36. *Figures 5 and 6: This figure should be simplified. You don't need to show the two years
separately, and I think you also don't need to show seasons separately. Thus you could
have just three subfigures (three levels) and the five regions inside of each one. Then use
a log scale for the AHTD, and symbols instead of bars (which will bring out the median
also more clearly).*
'We would like to show the simulation results separately because regional and temporal
impacts on the error were a part of the motivation for this study.'
This is not sufficient as a justification – the question is whether there are enough in-
teresting and relevant results to convey. As pointed out earlier, because of the small
sample size (just two times one starting date per season) this is questionable. Most of
the seasonal results are just representing general knowledge about seasonality of atmo-
spheric variability. I don't see a good reason for presenting the two years separately –
due to sample size, this is neither a good representation of year-to-year variability nor a
replacement for error bars. Also, the year-to-year variabilty is not of interest here. Make
your samples as large as possible, and then present one value for each!
'We added a horizontal line for the average error to all subfigures (see Figures 4 and 5).'
That is useful, but I would not call this 'average of the domain' but rather 'annual mean'
or 'average over all seasonal samples'.
'We did not use a log scale, because it would hide the seasonal and regional differences.'
It would not hide them, even though they would be less strongly visible. It is not a good
practice to present a set of figures where for the majority of them, more than 50% of the
graph area is unused. Also, the stratospheric results are hard to see at all. If you don't

want to use a log scale, at least you should optimise them and use different scales for each level.

37. *Figures 2, 3, 4, 7, 8: Please make sure that line width, colour intensity and marker size are sufficient to read all the content easily.*
'We tried to improve the figures accordingly'
I am sorry, but I fail to see real improvement. Lines are still tiny and for some, very pale colours are used. For example, the Heun line is hard to recognise in the plots.

38. 'This is a helpful evaluation and we added a statement in the paper summarizing the findings regarding the number of threads providing the minimum computation time with respect to the number of particles'.
Can you please point out where this statement is found? If you also find this useful, please make sure that it is properly presented.

39. OK

---

## Referee Report (RR2)

**Remarks on gmd-2016-314-author_response-version2.pdf**

1. Major comment 6: In response to my criticism of the wording 'total error', authors have changed that to 'global error'. They say: *Therefore, we changed the wording to global truncation error, which implies that the numerical errors investigated here originate from the accumulation of local truncation errors that are introduced at each time step. This terminology is consistent with the literature of numerical mathematics.*

   Unfortunately, even if this may be an established term in numerical mathematics, it still does not clearly convey the nature of the error which is analysed in this paper. As this is really a central topic of the whole work and its correct understanding is crucial, I think it is necessary to add a clear explanation at the first usage of the term. Furthermore, I find that the explanation provided in the response is not satisfactory. As I said in my review of the revised version, the transport deviation that we observe after 10 d is not only caused by truncation errors. Using different time steps also leads to different interpolation errors; these differences are amplified as well. Also, the final transport deviation is not just an 'accumulation of local truncation errors', such errors will be amplified in a way that is flow-dependent. I would therefore ask the authors to spend a full sentence or two on explaining what is behind the 'global error'.

2. Specific and Minor Remarks 1: The title. The authors now propose *Trajectory errors diagnosed with the MPTRAC advection module driven by ECMWF operational analyses*. While the term 'domain-specific' has been eliminated, my suggestion to include a reference to different numerical schemes and time steps was not taken into account. I leave it to the discretion of the editor how to handle that. However, I think that being more specific in the title will be very beneficial for the potential readership so that the content of the paper can be more quickly grasped, and including that still wouldn't make it too long.

3. Specific and Minor Remarks 3: 'Kinematic equations of motion'. The authors answer: *We traced back our usage of the phrase Equations of motion to the reference paper of Bowman et al. (2013). We also think this is a standard term in physics textbooks. We kept it as is but added the term trajectory equation as a synonym.* Bowman et al. is not a meteorological textbook (as requested). Physics textbooks would not be very relevant for a meteorological paper (and a specific reference hasn't been provided). Also, only in the third occurrence of the term (among many), 'or trajectory equation' has been added. As in meteorology the term 'equations of motion' is firmly connected to the dynamical equations (see the AMS Meteorological Glossary, `http://glossary.ametsoc.org/wiki/Equations_of_motion`), I find this misleading rather than helpful and I would ask the authors to stick to 'trajectory equation' (which they may explain as they feel proper, including to add a phrase such as 'sometimes also called kinematic equation of motion' if they deem that helpful).

All other issues are resolved in an acceptable way—thank you!

---

## Author Response (AR2)

**Reply to review comments**

Dear Dr. Pisso, dear Dr. Seibert,

we thank you for the time and efforts spent on the manuscript. We apologize that not all of the earlier comments have been addressed properly, but we hope that the revised draft solves the remaining issues. Please find our point-by-point replies below (colored in blue).

**Reviewer #1**

**0.1  General**

The authors have implemented suggestions and thus improved their paper. However, not all recommendations have been sufficiently addressed, as detailed below. Specifically, certain problems related to meteorological reasoning, for example invoking turbulence as explanation for larger transport errors, have not been resolved.

Quotations from my original comments are set in *italics*, those from the authors' answers or revised manuscript in 'single quotes'.

**0.2  Major remarks**

1. clarified

2. *Another open question is whether RK4 with 60 s time step is a suitable reference method.*
The authors accept this critism for the northern midlatitudes. However, they don't offer an improvement or a justification of keeping their reference. I would expect that the authors address this point either by repeating the calculations and evaluations with a shorter reference time step, or by adding an explanation and justification within their paper. Their remark on Hoffman et al. (2016) is not leading to an argument, it just repeats and confirms what I wrote.

In order to better justify our choice, we used a time step of 30 s to recompute the reference trajectories for January 2015. The resulting absolute horizontal transport deviations after 10 days changed by 0-3.3 km in the stratosphere, by 0.6-8.5 km in the UT/LS region and by 4-14 km in the troposphere in this test case. These differences are much smaller than the AHTDs presented in Fig. 5 and we can therefore consider our results to be robust. The only exception would be the high and mid latitudes of the southern hemisphere of the stratosphere, but here the numerical errors are very small anyway (AHTD <6 km).

We think that a reference calculation with a 60 s time step is adequate for all presented cases. We changed the wording to make clear that by convergence, the convergence of the transport deviations relative to simulations with larger time steps is meant:

Sensitivity tests using variable time steps down to 30 s showed that the numerical solution from the RK4 method converges for time steps of 60 s or less, in the sense that transport deviations relative to simulations with a time step of 120 s do not change significantly.

The remark to Hoffmann et al. (2016b) was made to clarify that the two studies are not contradicting each other, because the specific region (UTLS) needs to be taken into account when comparing the results of the studies.

3. clarified

4. OK.

5. *Finally, the results are certainly sensitive to the resolution advection module and ECMWF operational analyses of the wind field data. Results obtained for the specific case of 16 km / 3 h therefore cannot be generalised. Keeping in mind the conclusions of Stohl et al. (1995), Brioude et al. (2012), and Bowman et al. (2013), 3 h intervals for the wind fields are coarser than what would be desired at this horizontal resolution. As 1 h is provided by ECMWF, I am wondering why it was not used.*
I don't feel satisfied by the answer given. It is not appropriate to call the 3 h 'officially approved resolution' (there is no such thing as official approval, all what is in MARS is usable). The question of how results would change with 1 h data is highly pertinent.
Alternatively, one could test coarser temporal in combination with coarser spatial resolution.

We agree that a study of truncation errors with a high temporal resolution, which fits better to the fine horizontal resolution, would be interesting. We admit that the 3 h time interval is a potential weakness of this study. Considering that substantially more ECMWF data would have to be downloaded, all simulations would have to be repeated and the complete paper would have to be rewritten, we think that this refinement is not feasible now. We discuss this potential weakness of the study in the concluding section.
A comparison of different resolutions is beyond the scope of this study. We already vary the region, integration method and time step, which are quite many variables for one study.

6. New issue created by introducing the wording 'total error' in recognition of the fact that the transport errors obtained are not pure truncation errors: As shortening the time step eliminates only errors introduced by the truncation error, the 10-day error is not the total error. The total error would be larger than that as it has other contributions as well which are also amplified during the 10-day transport. The authors should make that clear and find another, more appropriate wording. See also major remark #1 of Reviewer #2.

We agree that the term *total error* is misleading, it would include other error sources

that are not considered in this study. Therefore, we changed the wording to *global truncation error*, which implies that the numerical errors investigated here originate from the accumulation of local truncation errors that are introduced at each time step. This terminology is consistent with the literature of numerical mathematics.

**0.3  Specific and Minor Remarks**

1. Authors now propose the title 'Domain specific trajectory errors diagnosed with the MPTRAC advection module and ECMWF operational analyses'. First of all, one would have to hyphenate 'domain-specific'. Then, I think the word *domain* is not the best choice to express that evaluations were done separately for different regions (*domain* usually refers to a calculation domain and not to a climatological region.) Furthermore, MPTRAC and ECMWF data are not on the same level, one is the model, the other model input. Better say something like 'Trajectory errors as a function of numerical scheme, time step, and region of the atmosphere diagnosed with the MPTRAC model'. Maybe you can add 'for ECMWF wind fields', but I think it is not so important to bring that into the title.

We changed the term 'domain' to 'region' throughout the manuscript and removed it from the title. We also rephrased the title to clarify that the calculations are driven by external ECMWF operational data:

Trajectory errors diagnosed with the MPTRAC advection module driven by ECMWF operational analyses

2. Page 1, line 1: Abstract. OK

3. *Page 1, line 4: kinematic equation of motion (comes also in other places). I don't feel comfortable with this wording. Equations of motion for me would refer to the Euler or Navier-Stokes equations. Why not call this the trajectory equation?* Authors decline to change their wording without providing arguments. I will accept their wording if they provide a quotation from a well-established meteorological textbook which uses 'equation of motion' for the kinematic trajectory equation.

We traced back our usage of the phrase Equations of motion to the reference paper of Bowman et al. (2013). We also think this is a standard term in physics textbooks. We kept it as is but added the term *trajectory equation* as a synonym.

4. OK

5. It is not true that no forecasts are produced from 06 and 18 UTC analysis. However, these are not long-term forecasts but just for providing background fields at the next major analysis step.
I accept the argument that these details don't belong to the introduction. Therefore, I would suggest to provide all the information about resolution and kind of ECMWF input fields only in the Section 2, and to remove them completely from Section 1. Otherwise,

one is wondering about partial information until one has reached the next section.

We removed information regarding the time resolution from the introduction to avoid confusion for the reader at this point. The horizontal resolution is still mentioned to show that relatively fine resolved meteorological data are used. The temporal resolution and information about the data processing are now given in Section 2:

For usage with MPTRAC, the wind fields were retrieved on model levels with a longitude-latitude grid with $0.125° \times 0.125°$ resolution and have been interpolated vertically to 114 pressure levels in the troposphere and stratosphere up to 5 hPa with help of the Climate Data Operators (CDO, 2015). 12-hourly analyses were combined with short-term forecasts in between to obtain data with a 3-hourly time step.

6. OK

7. Reformulation is ok. I don't have the impression that Heng et al. (2016) provides a full model description.

This paper focuses on the advection module only, therefore we think no full model description is necessary. However, we added references to Heng et al. (2016) and Hoffmann et al. (2017) for readers that are interested in more information about the model.

8. OK

9. The explanation is ok but it should also go into the revised manuscript.

We added information about the data conversion to the manuscript, see answer to minor remark #5.

10. OK

11. *Page 6, line 4 ff.: k1 = . . . .It seems that you define certain velocities as k.* I regret that the authors want to keep this notation. In any case, symbols have to be explicitly explained (in words) before or immediately after first usage, whether or not there is an equation which defines them. This is a standard in scientific publications.

We added an explanatory phrase between Eq. (7) and (8).

12. OK

13. OK

14. *Page 11, line 30: The tropospheric mid-latitudes were expected to cause the largest errors, because the most complex wind systems occur in this region due to a larger land surface ratio and more complex orography.* The authors offer some improvement, but it is still insufficient.
(1) They should pay attention to their style. For example, wordings such as 'The troposphere has its largest error' are not correct (this example is not the only such mistake). The troposphere cannot have any error, only calculations can.

(1) We tried to improve the wording in the whole paragraph.

(2) We still find the phrase 'the evolution of northern mid-latitudes meteorological systems is more difficult to simulate than for the southern mid-latitudes due to the larger land-sea ratio and more complex orography of the northern hemisphere.' Apart from the question whether this is true or not, the difficulty to simulate the evolution of meteorological systems (in other words, the predictability) is not relevant for trajectory errors based on analyses.

(2) Our intention was to give reasons for larger trajectory errors in the northern mid-latitudes compared to the southern mid-latitudes. We changed our statement accordingly:

In addition, stronger fluctuations are expected in the northern mid-latitudes compared to the southern mid-latitudes due to the larger land-sea ratio and more complex orography of the northern hemisphere.

(3) 'These errors are caused by . . . and higher turbulence in the underlying region'. In their answer to minor remark 16 authors admit that turbulence is not relevant thus, why do they again come up with turbulence as an explanation of trajectory errors?

(3) We used the term turbulence for small-scale fluctuations of the wind field, which was wrong. We changed it throughout the manuscript.

15. OK

16. *Page 12, line 5: The relative high errors in the tropics are probably caused by a stronger turbulence in that region. The lower bound of the stratospheric region of our test cases is 16 km, since the tropopause reaches an average altitude of 16 km near the ITCZ, turbulent movements due to deep convection can occur more frequently in the lower stratosphere above the tropics.*
The authors admit that the term 'turbulence' is misleading here, but they have not changed their wording. The explanation by turbulence is wrong and has to be removed. If they want to refer to 'fluctuations in the meteorological input data', I think they have to be more specific what is different in the tropics compared to mid-latitudes. Mid-latitude wind fields also show fluctuations.

Our original explanations was misleading. We would like to explain why trajectory errors in the tropical stratosphere are larger than in other stratospheric regions and changed the paragraph to:

The errors of the simulations in the stratosphere are typically below 25 km, except for January 2015. Stratospheric trajectory errors in the tropics are larger than in the other regions, which is probably due to the close vicinity of the tropical tropopause, which reaches an average altitude of 16 km near the ITCZ.

17. OK

18. *Page 12, line 12: We need to stress that each simulation lasts only 10 days, which*

*is a relatively short time interval to analyze seasonal effects. Fast temporal variations and changes in medium-range weather patterns can blur out the impact of seasons that is observed here. To better resolve the seasons you don't need longer trajectories, but more frequent starts or more years. I any case, I don't think that the seasonal effects are so interesting, you could discuss this just briefly. It is obvious that stronger variations in the wind fields will lead to larger truncation errors, and the dependence of the variability of wind fields on the seasons is well known.*

Most of this comment is not at all addressed in the reply by the authors. It is really questionable whether seasons are represented in a statistically adequate way with a single day on which trajectories were started. Could you not just add some more? They did 5000 (10-day) trajectory simulations, with a parallelised model on a supercomputer. Doing 20000 or 50000 instead would not be a serious burden in terms of computing work. It is true that the section on seasonal results is not very long, but if we add also the figures, it is also not so short. Generally speaking, I consider the merits of this paper lying in the realm of numerical methods, showing a systematic comparison of a number of numerical integration schemes. The layer of meteorological interpretation according to season, region etc. which has been put around that has much less scientific substance, and in its present formulation even is partly mistaken (see the turbulence issue). It would be better to de-emphasise this part.

Our discussion of seasonal and year-to-year variability has already been shortened and we try to make clear that it is not meant to be statistically representative. However, Figure 5 of the manuscript clearly indicates that some seasonal effects are present and that year-to-year variability might be large. We do not want to average the few data points of each region, because then possible effects of seasons or year-to-year variability would be completely hidden. We can not claim these effects to be statistically significantly due to the limited sample size, but we think that they are relevant and want to underline that trajectory errors depend on many factors. As the average of the seasonal samples has been added to the figures following earlier review comments, we think that it does no harm to show all available information.

Each of the 5,000 simulations uses 500,000 trajectories which results in high computation costs. A study of seasonal and year-to-year variations would be much easier and cheaper if only one integration method and one time step would be used. If more representative results are necessary, a separate study could be conducted in the future.

19. OK

20. *Page 12, line 35: The median error gets somewhat larger in the troposphere, where particle paths are more likely being affected by atmospheric turbulence. Hoffmann et al. (2016a) says that MPTRAC uses the same diffusivity throughout troposphere and stratosphere. How is this compatible?*

The authors say that they answered this in their response to major comment 1. However, there they only explain that turbulence is not active in their simulations. But then, this argument simple collapses; however, the sentence in question has been left unchanged.

We were misleadingly referring to turbulence instead of small-scale fluctuations of the wind field. We changed the sentence to:

The median error is somewhat larger for simulations in the troposphere, where particle paths are more likely being affected by synoptic-scale fluctuations of the wind field.

21. *It would be useful to explain why you are only testing OpenMP and a single node if MPTRAC is capable to work on distributed-memory systems as well.* Answer: 'We added the following sentence in Sect. 3.4: The MPI parallelization is only used for ensemble simulations, which are conducted independently on multiple nodes. Therefore, the scalability of the MPI parallelization is mostly limited by I/O issues, which are out of scope of this study.' I don't understand the argument. If the strategy for distributed-memory machines is trivial parallelisation by multiple runs started concurrently, why does the code offer MPIbased parallelisation?

In our computing environment it is easier to conduct ensemble simulations (e.g. Heng et al. (2016)) if the code offers MPI parallelization.

22. The explanation 'The time measurements refer only to the part of the code spent in the advection module of MPTRAC.' should be included also in the manuscript text. Furthermore, if this is the case, the contribution independent of the number of threads also stems from non-parallel parts of the code, not only from the OpenMP overheads.

We added the information that only the advection module is analyzed (as the other physics modules are not enabled). The advection module does not have a non-parallel part. The code structure shows that time measurement directly embraces the parallel section:

```
START_TIMER(timer_phys);
**pragma omp parallel for default(shared) private(dt,ip)**
for (ip = 0; ip < atm->np; ip++)
 module_advection(ip, ...);
STOP_TIMER(timer_phys);
```

23. The explanation given should be included in the manuscript text.

We added the explanation to the manuscript:

For smaller number of particles ($10^4$ or less) the speedup is limited by the overhead of the OpenMP parallelization and by load imbalances, which can also become significant for larger numbers of parcels if SMT is not enabled for all cores jointly.

24. As the Pettersson method is widely used, it would really be desirable that the authors test its efficiency with a reasonable iteration cut-off compared for example to the midpoint method.

As it is difficult to define a reasonable iteration cut-off, we executed new simulations with an implementation of the Petterssen method that uses exactly two iterations. The accuracy in most simulations is similar or slightly worse than the accuracy of the midpoint

method. In conclusion, none of our three implementations of the Petterssen scheme was more efficient than the midpoint method or the Runge-Kutta 3 method.

The accuracy of the Petterssen scheme with one iteration (Heun's method) is somewhat worse than the midpoint method. When two iterations of the Petterssen scheme are computed, the transport deviations are closer to those obtained with the midpoint method. The best efficiency, which we define as lowest computational costs when adhering to our error limit, is mostly obtained with the midpoint and RK3 methods. The additional iterations of the Petterssen scheme improve the accuracy, but they are too computational expensive for our model. In general, a well defined convergence limit for the number of iterations is needed for an efficient application.

25. It would be good to put the explanations, e.g. about the role of cache for different numerical schemes, into the manuscript text.

We added a note on the influence of the hardware:

Note that the hardware, especially the memory cache, affects the six integration schemes differently. A single call to the wind interpolation function is up to 50% cheaper for a higher order method compared to Euler's method, because the cache is used more efficiently.

26. OK

27. OK

28. *Page 14, line 25: We attribute this to larger small-scale variations caused by atmospheric turbulence and mixing in the troposphere. The first part of the explanation is correct, but the second part not. These variations are not caused by turbulence (16 km is not turbulence scale !!) and certainly not by mixing (this would reduce and not amplify variability!).*
The authors have removed the reference to mixing, but they keep the reference to turbulence. As I have tried to explain in various parts of the paper where turbulence is invoked, this is not appropriate. Atmospheric turbulence does not create variability at the high-frequency end of the resolved motion scales. It will rather tend to undo existing gradients.

We corrected this and removed the reference to turbulence.

29. Summary and conclusions: Please spell out RK where it occurs for the first time in this section (some people may read only this section).

The first occurrence of Runge-Kutta has been spelled out in the conclusions.

'After 24 h the trajectory errors are quite similar in the troposphere and stratosphere' Figure 4 shows a difference by about a factor of 10 for AHTD, thus they are not 'quite similar'.

We agree that the difference is relevant and changed the sentence accordingly.

The trajectory errors after 24 h are shown, as they are expected to be less affected by individual meteorological conditions than the errors after 10 days. The errors of the higher order integration schemes with a time step of 120 s are in the order of 80-200 m in the troposphere. The errors of the simulations in the stratosphere are about ten times smaller and the discrepancy between the error in the troposphere and stratosphere becomes even larger to a factor of about 25 when the global truncation errors after ten days are analyzed.

When analyzing the errors after 24 h simulation time we noticed that there was a bug in the plot script, resulting in errors that are five times too large. Only the curves for the AHTDs after 24 h were affected in the bug. We corrected the figure and the paragraph about the suggested time step.

After 24 h, when trajectory errors are mostly influenced by truncation errors, the diffusivity-based error limit is not particularly strict, which allows us to use large time steps for the calculations. In fact, even the results obtained with the longest time step of 1 h adhere to the error limit for the higher order methods as shown in Fig. 8.

'We attribute this to larger small-scale variations caused by atmospheric turbulence.' Remove erroneous reference to turbulence.

The erroneous reference was corrected.

Statistics not being sufficiently robust: as said before, it would be desirable to increase the sample size.

More simulations could be executed in another study, after the decision for an integration method and time step was made, which was the aim of this study.

30. *Page 15, line 79: The study of Seibert (1993) . . . . To achieve truncation errors that are smaller than overall trajectory uncertainty, they found that the time step should fulfill the CFL criterion as a necessary condition for convergence. The recommendation there for a sufficiently small truncation error was 15% of the time step needed for convergence of the Petterssen scheme. If we assume that the reference accuracy has also improved in the meantime, an even smaller value would result. The CFL criterion is recommended to make sure that no small-scale features are skipped, not for convergence of the iterations in the Petterssen scheme.* The authors have amended their wording, but they have not changed the first sentence quoted above, which is not an accurate representation of Seibert (1993), as explained in my previous comment.

We apologize for the wrong representation of your conclusions. We decided to omit this paragraph as we are still unsure if we fully understand the statement. The paragraph was replaced by a more general discussion of the used time interval of 3 h as mentioned in the answer to major remark #5.

31. OK

32. OK

33. 'For better visibility of the circulation patterns we decided to use different scales for the three pressure levels. In case of vertical velocities the maximum values differ by more than a magnitude between the levels.' I think a part of the motivation for showing sample vertical motion patterns at three different levels is exactly this fact that vertical motions are much smaller in the stratosphere, and this is hidden by changing the scale between the figures. 'The chosen pressure levels are used in the model and correspond closely to the altitudes given in the figure caption.' If standard pressure levels also are available, it would be preferable to shown them. 'The colour coding for vertical velocity has been reversed following comments by reviewer #2.' Thank you.

Unfortunately, the data are only available on model levels and not on standard pressure levels. We added a note to the caption to remind the reader that the scales vary.

34. OK

35. I hope this annotation will really happen  why not doing it now?

Annotations will be added if they are requested during the formatting phase.

36. *Figures 5 and 6: This figure should be simplified. You don't need to show the two years separately, and I think you also don't need to show seasons separately. Thus you could have just three subfigures (three levels) and the five regions inside of each one. Then use a log scale for the AHTD, and symbols instead of bars (which will bring out the median also more clearly).*
'We would like to show the simulation results separately because regional and temporal impacts on the error were a part of the motivation for this study.' This is not sufficient as a justification  the question is whether there are enough interesting and relevant results to convey. As pointed out earlier, because of the small sample size (just two times one starting date per season) this is questionable. Most of the seasonal results are just representing general knowledge about seasonality of atmospheric variability. I don't see a good reason for presenting the two years separately  due to sample size, this is neither a good representation of year-to-year variability nor a replacement for error bars. Also, the year-to-year variability is not of interest here. Make your samples as large as possible, and then present one value for each!

We tried to summarize our reasons to show all samples in reply to comment #18.

'We added a horizontal line for the average error to all subfigures (see Figures 4 and 5).' That is useful, but I would not call this 'average of the domain' but rather 'annual mean' or 'average over all seasonal samples'.

We followed the suggestion:

The errors obtained in the polar regions are second largest with an average over all seasonal samples of around 200 km and peak errors in polar summer of up to 380 km.

'We did not use a log scale, because it would hide the seasonal and regional differences.' It would not hide them, even though they would be less strongly visible. It is not a good

practice to present a set of figures where for the majority of them, more than 50% of the graph area is unused. Also, the stratospheric results are hard to see at all. If you don't want to use a log scale, at least you should optimise them and use different scales for each level.

We used identical scales to underline the differences between the regions, but we agree that readability is more important and optimized the scale for each subfigure. We think that the very small median values as depicted in Fig. 5 do not make it necessary to provide actual values to the reader. Therefore, we decided to keep the linear scale.

37. *Figures 2, 3, 4, 7, 8: Please make sure that line width, colour intensity and marker size are sufficient to read all the content easily.* 'We tried to improve the figures accordingly' I am sorry, but I fail to see real improvement. Lines are still tiny and for some, very pale colours are used. For example, the Heun line is hard to recognise in the plots.

We increased the linewidth further which also reduced the problem with pale colors.

38. 'This is a helpful evaluation and we added a statement in the paper summarizing the findings regarding the number of threads providing the minimum computation time with respect to the number of particles'. Can you please point out where this statement is found? If you also find this useful, please make sure that it is properly presented.

We included a paragraph about the most efficient configuration for different numbers of parcels:

The fastest simulations for a set of about $10^2$ parcels are possible with 4 cores. 12 cores should be used when $10^3$ parcels are simulated. For $10^4$ parcels the simulations with 24 cores are fastest. For $10^5$ or more parcels all 48 cores (which includes SMT) should be used.

39. OK

[revised manuscript text omitted]

---

## Author Response (AR3)

**Reply to review comments**

We thank the reviewer for the time and efforts spent on the manuscript. We considered the remaining comments and hope that the revised draft properly addresses the open issues. Please find our point-by-point replies below (colored in blue).

**Reviewer #1**

1. Major comment 6: In response to my criticism of the wording total error, authors have changed that to global error. They say: *Therefore, we changed the wording to global truncation error, which implies that the numerical errors investigated here originate from the accumulation of local truncation errors that are introduced at each time step. This terminology is consistent with the literature of numerical mathematics.*

Unfortunately, even if this may be an established term in numerical mathematics, it still does not clearly convey the nature of the error which is analysed in this paper. As this is really a central topic of the whole work and its correct understanding is crucial, I think it is necessary to add a clear explanation at the first usage of the term. Furthermore, I find that the explanation provided in the response is not satisfactory. As I said in my review of the revised version, the transport deviation that we observe after 10 d is not only caused by truncation errors. Using different time steps also leads to different interpolation errors; these differences are amplified as well. Also, the final transport deviation is not just an accumulation of local truncation errors, such errors will be amplified in a way that is flow-dependent. I would therefore ask the authors to spend a full sentence or two on explaining what is behind the global error.

To clarify we added "However, note that the final transport deviation is not just an accumulation of local truncation errors, because local errors will be amplified in a way that is flow-dependent. The dependency on the flow may cause significant variability in the global truncation errors, in particular if the integration covers time periods of several days." in the introduction section close to where terms are introduced.

2. Specific and Minor Remarks 1: The title. The authors now propose *Trajectory errors diagnosed with the MPTRAC advection module driven by ECMWF operational analyses.* While the term domain-specific has been eliminated, my suggestion to include a reference to different numerical schemes and time steps was not taken into account. I leave it to the discretion of the editor how to handle that. However, I think that being more specific in the title will be very beneficial for the potential readership so that the content of the paper can be more quickly grasped, and including that still wouldnt make it too long.

In the revised manuscript we changed the title to "Trajectory errors of different numerical integration schemes diagnosed with the MPTRAC advection module driven by ECMWF operational analyses". Adding a reference to the different time steps seems to

make the title a bit long and is perhaps already addressed indirectly by referring to the different numerical schemes? However, we would be fine to change the title once more if this is considered to be more clear by the editor.

3. Specific and Minor Remarks 3: Kinematic equations of motion. The authors answer: *We traced back our usage of the phrase Equations of motion to the reference paper of Bowman et al. (2013). We also think this is a standard term in physics textbooks. We kept it as is but added the term trajectory equation as a synonym.* Bowman et al. is not a meteorological textbook (as requested). Physics textbooks would not be very relevant for a meteorological paper (and a specific reference hasnt been provided). Also, only in the third occurrence of the term (among many), or trajectory equation has been added. As in meteorology the term equations of motion is firmly connected to the dynamical equations (see the AMS Meteorological Glossary, `http://glossary.ametsoc.org/wiki/Equations_of_motion`), I find this misleading rather than helpful and I would ask the authors to stick to trajectory equation (which they may explain as they feel proper, including to add a phrase such as sometimes also called kinematic equation of motion if they deem that helpful).

We rephrased "kinematic equation of motion" by "trajectory equation" throughout the manuscript.

All other issues are resolved in an acceptable way – thank you!

Thank you again for providing feedback!

[revised manuscript text omitted]